

# Exploring aerosol-cloud interactions in liquid-phase clouds over eastern China and its adjacent ocean using the WRF-Chem-SBM model

Jianqi Zhao[1], Xiaoyan Ma[1], Johannes Quaas[2], Hailing Jia[3]

[1]Key Laboratory for Aerosol-Cloud-Precipitation of China Meteorological Administration, Nanjing University of Information Science & Technology, Nanjing 210044, China
[2]Institute for Meteorology, Leipzig University, Leipzig, Germany
[3]SRON Netherlands Institute for Space Research, Niels Bohrweg 4, 2333 CA Leiden, the Netherlands

*Correspondence to*: Xiaoyan Ma (xma@nuist.edu.cn)

**Abstract.** This study aims to explore aerosol-cloud interactions in liquid-phase clouds over eastern China (EC) and its adjacent ocean (ECO) in winter based on WRF-Chem-SBM model which couples a spectral-bin cloud microphysics (SBM) and online aerosol module (MOSAIC) as well as the four-dimensional assimilation approach. The model evaluation demonstrates that assimilation has a predominantly favorable impact on the simulation, and the model reasonably reproduces the satellite-retrieved cloud parameters. Differences in meteorological, topographic and aerosol conditions lead to differences in EC and

ECO aerosol-cloud processes. Multiple atmospheric supersaturation pathways and abundant aerosols in EC enable more aerosols to be activated, but limited water content makes them difficult to grow into large droplets. While atmospheric supersaturation pathway and aerosol number concentration ($N_{aero}$) limit the cloud droplet number concentration ($N_d$) in ECO, the relatively abundant water content enables more large cloud droplets to exist here. EC and ECO cloud liquid water content (CLWC) exhibit close variation trends with $N_d$, but differences in aerosols, supersaturation pathways, and water vapor

conditions result in distinctions in CLWC variations and the differentiation between precipitation and non-precipitation clouds in the two regions. Meteorological conditions suitable for EC cloud development include (1) weak winds and strong surface radiative forcing cooling, (2) moist air brought by strong easterly winds, (3) cooling and topographic uplift caused by strong northerly winds, and (4) strong updrafts. Meteorological conditions suitable for ECO cloud development include (1) aerosol-rich and not excessively dry airflow from moderate westerly wind, (2) cooling caused by northerly winds, and (3) updrafts. In

general, the effect of cooling on aerosol activation is more pronounced compared to humidification in EC and ECO. Moreover, the aerosol conditions suitable for aerosol activation and CLWC increase are similar overall, with the difference being that aerosol activation is strongest under moderate $N_{aero}$ conditions, whereas high CLWC to $N_{aero}$ ratios are often seen under low $N_{aero}$ conditions in addition to moderate $N_{aero}$ conditions.

## 1 Introduction

Atmospheric aerosols have significant effects on the Earth's radiation balance, water cycle, and climate system through direct



absorption and scattering of solar radiation as well as indirect effects on cloud formation and development by acting as cloud condensation nuclei (CCN) and ice nuclei (IN) (Carslaw et al., 2010; Wilcox et al., 2013; Tian et al., 2021). The latter, known as the aerosol indirect effect, or more recently by the Intergovernmental Panel on Climate Change (2013) defined as effective radiative forcing due to aerosol-cloud interactions, $RF_{aci}$, remains a challenging scientific topic in climate assessment and prediction because of its complex mechanisms and high uncertainties (Church et al., 2013; Jia et al., 2019b; Arias et al., 2021). Liquid-phase clouds offer great opportunities to untangle aerosol indirect effect due to their sheer abundance and impact on cloud radiative forcing (Christensen et al., 2016).

Twomey (1977) pointed out that under a constant cloud water content, the activation of atmospheric aerosol particles entering into clouds leads to an increase in cloud droplet number concentration ($N_d$), a decrease in droplet size, and an increase in cloud albedo. This mechanism, termed the aerosol first indirect effect, is revealed to be the key driver of aerosol indirect effect, besides, the rapid adjustments also contribute significantly (Quaas et al., 2020). Two key competing mechanisms exist in the latter, one of which is that an increase in $N_d$ causes a decrease in precipitation efficiency and with this, a co-increase in cloud liquid water path (CLWP) and cloud fraction (CF), this mechanism dominates in precipitation clouds (Albrecht, 1989). The other mechanism dominates in non-precipitating clouds, i.e., with limited water content, the decrease in droplet size reduces sedimentation velocity and increases cloud-top liquid water content, resulting in additional cloud top cooling and pushing further entrainment and evaporation (Bretherton et al., 2007). Moreover, as cloud droplets decrease in size, their ratio of surface area to volume is higher and evaporation is faster, resulting in further enhancement of the negative buoyancy at cloud top (Small et al., 2009). Numerous studies have been conducted to assess the contribution of these three mechanisms. Statistical analysis based on satellite-retrieved data indicates that the CLWP of marine low clouds exhibits a weak decreasing trend with rising $N_d$ caused by aerosol increase (Michibata et al., 2016; Rosenfeld et al., 2019). Gryspeerdt et al. (2019) found that CLWP is positively correlated with $N_d$ at low $N_d$ and droplet size greater than the precipitation threshold, i.e., delayed precipitation leads to increased CLWP. In contrast, for the clouds with high $N_d$ and low possibility of precipitation, CLWP shows a negative correlation with $N_d$. In this case, the increase of aerosol leads to the decrease of cloud droplet size and the increase of $N_d$, which in turn accelerates the mixing and evaporation process and makes CLWP decrease. The CLWP response to aerosols differs clearly between precipitation and non-precipitation clouds because of the significant influence of precipitation process on CLWP (Christensen and Stephens, 2012). CLWP has a significant positive correlation with the aerosol index (AI) in precipitation clouds, and the opposite in non-precipitation clouds (Chen et al., 2014). Furthermore, the response of CLWP to aerosol highly depends on meteorological conditions. Chen et al. (2014) indicated that CLWP and aerosol concentration show a negative correlation when entrainment mixing exerts a marked impact on the cloud-side evaporation process (which usually occurs under free troposphere with dry and unstable atmosphere), and this relationship shifts to positive as the atmosphere becomes moist and stable. Such statistical analysis, however, suffers severely from retrieval uncertainties (Arola et al., 2022). In turn, also "opportunistic experiments" such as the analysis of ship and pollution tracks hint at a decrease in CLWP but an increase in cloud horizontal extent in response to aerosol increases (Toll et al., 2019; Christensen et al., 2022). In spite of considerable efforts in recent researches to unravel aerosol-cloud interactions, it remains challenging to distinguish



and quantify underlying mechanisms of aerosol-cloud interactions under diverse air pollution and meteorological conditions.

In order to further resolve the mechanisms of aerosol-cloud interactions, the proper use of numerical simulations is necessary. Current global climate models (GCMs) have difficulties in accurately representing the response of cloud to aerosol, which is mainly due to (1) the limitation of coarse model resolution, (2) the absence of sufficient consideration of cloud droplet spectral characteristics, and (3) the fact that most current GCMs parameterize the precipitation mechanism through the

autoconversion process as an inverse function of $N_d$, without accurate representation of entrainment-mixing processes (Quaas et al., 2009; Bangert et al., 2011; Michibata et al., 2016; Zhou and Penner, 2017). Regional climate models (RCMs) with higher resolution and finer physical parameterization can effectively compensate for at least some of these shortcomings and better reproduce the physical processes, which help to further distinguish and quantify the aerosol-cloud interaction mechanisms (Li et al., 2008; Bao et al., 2015). The Weather Research and Forecasting model (WRF) has been widely used in regional numerical

simulation studies because of its advanced technology in numerical calculation, model framework, and program optimization, which has many advantages in portability, maintenance, expandability, and efficiency (Maussion et al., 2011; Islam et al., 2015; Xu et al., 2021). The chemistry-coupled version of the WRF model (WRF-Chem) allows to simulate the spatial and temporal distributions of reactive gases and aerosol, spatial transport and their interconversion while simulating meteorological fields and atmospheric physical processes (Tuccella et al., 2012; Sicard et al., 2021). Bulk and bin approaches are commonly

utilized to simulate regional cloud microphysical processes. Bulk schemes diagnose the size distribution of hydrometeor based on different predicted bulk mass (one-moment schemes) or number and mass mixing ratios (double-moment schemes) and assumed size distribution, showing significant limitations in reproducing processes such as condensation, deposition and evaporation (Lebo et al., 2012; Wang et al., 2013; Fan et al., 2015). Bin schemes predict the size distribution of hydrometeors based on a number of discrete bins, enabling better representation of cloud microphysical processes. As stated by Khain et al.

(2015), previous studies have demonstrated that bin schemes outperform bulk schemes in simulations. In this study, the WRF-Chem-SBM model (Gao et al., 2016) is used, in which the Model for Simulating Aerosol Interactions and Chemistry (MOSAIC) in WRF-Chem (Fast et al., 2006) is coupled with a spectral-bin microphysics (SBM) scheme (Khain et al., 2004). In WRF-Chem-SBM, aerosol information is provided for cloud microphysical simulations, and cloud microphysical parameters are offered to aerosol-chemistry simulations, which are of great help to reproduce accurate aerosol and cloud conditions as well

as to distinguish and quantify aerosol-cloud interaction mechanisms.

Eastern China (EC) is one of the most human-active regions worldwide, resulting in numerous anthropogenic aerosol emissions. The contrast between the high aerosol-content air masses of EC and the relatively clean air masses of the Pacific Ocean makes EC and its adjacent ocean (ECO) ideal regions for exploring aerosol-cloud interactions (Fan et al., 2012; Wang et al., 2015; Zhang et al., 2021). It is shown that low clouds contribute the most to the Earth's energy balance due to their broad

coverage and the albedo effect governing their impact on emitted thermal radiation (Hartmann et al., 1992). The statistics of Niu et al. (2022) using the satellite data from 2007-2016 show that low clouds in EC and ECO occur most frequently in winter, reaching more than 50%, with stratocumulus clouds, which are persistent and sensitive to aerosol variations (Jia et al., 2019a), constituting more than 70% of the low clouds. Therefore, the EC and ECO aerosol-cloud response in winter is an ideal



condition to investigate aerosol-cloud interactions in liquid-phase clouds. Based on the WRF-Chem-SBM model, we
investigate the aerosol-cloud interaction mechanisms of EC and ECO in winter by obtaining detailed and high-resolution
aerosol, cloud parameters as well as meteorological information through reproduction of real scenarios.

The paper is structured as follows: Section 2 introduces the model configuration and observational data used in the study,
Section 3 presents the evaluation of simulated results and the analysis of aerosol-cloud responses presented in the simulations,
and the summary is given in Section 4.

## 2 Methods and Data

### 2.1 Simulation Setup

We performed model simulations using the WRF-Chem-SBM (Gao et al., 2016). In this model, the 4-bin MOSAIC aerosol
module treats mass and number of nine major aerosol species, including sulfate, nitrate, sodium, chloride, ammonium, black
carbon, primary organics, other inorganics, and liquid water (Zaveri et al., 2008). The diameters of 4 bins ranges from 0.039-
0.156, 0.156-0.624, 0.624-2.5 and 2.5-10.0 μm, respectively, and aerosol particles are assumed to be internally mixed. This
module is capable of treating processes such as emissions, new particle formation, particle growth/shrinkage due to uptake/loss
of trace gases, coagulation, dry and wet deposition (Sha et al., 2019). In addition, this model incorporates the fast version of
SBM, which solves a system of prognostic equations for three hydrometeor types (liquid drops, ice/snow and graupel) and
CCN size distribution functions (Khain et al., 2010). Each size distribution is structured by 33 mass doubling bins (i.e., the
mass of the particle in the kth bin is twice that of the k-1th bin). The cloud microphysical processes described in the SBM
contain aerosol activation, freezing, melting, diffusion growth/evaporation of liquid drops, deposition/sublimation of ice
particles, drop and ice collisions.

The model domain is shown in Fig. 1, where two sets of two-layer nested grids are employed. The outer grids (12 km
resolution) have centroids and grid points of (32°N, 120°E) and 151 × 125, while the inner grids (4 km resolution) represent
EC (160 × 160 grid points) and ECO (121 × 121 grid points), respectively. There are 48 vertical layers up to 50 hPa, with layer
spacing extending from 40 m near the surface to 200 m at 3000 m altitude and over 1000 m above 10000 m altitude. The
simulations run from 00:00:00 UTC on 1 Feb 2019 to 00:00:00 UTC on 13 Feb 2019, where the first 24 h are disregarded as
spin-up and not involved in subsequent analyses. Meteorological initial and boundary conditions are obtained from the
National Center for Environmental Prediction (NCEP) FNL global reanalysis data with 0.25° resolution and available every 6
h (https://rda.ucar.edu/datasets/ds083.3, last access: 11 October 2023), and anthropogenic emission sources come from the
Multi-resolution Emission Inventory for China (MEIC) 2016 version developed by Tsinghua University
(http://meicmodel.org.cn, last access: 19 March 2023). As presented in Fig.1, the anthropogenic aerosols of EC and ECO  are
dominated by EC under winter monsoon, although the model domain contains countries and regions other than China, MEIC
can satisfy the anthropogenic aerosol simulation of the region concerned in this study. The model parameterization settings
are listed in Table 1.





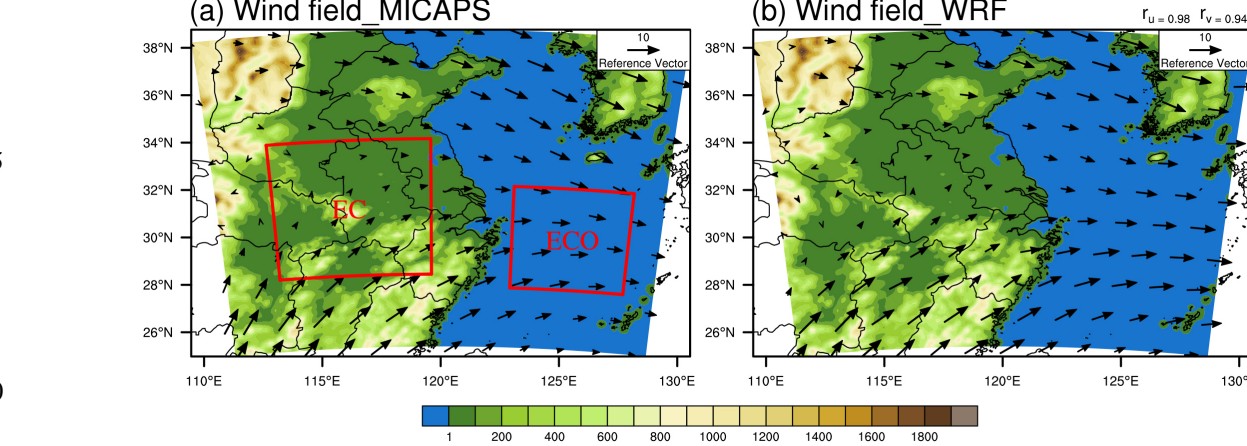

**Figure 1.** Topography (unit: m) of the model domain, MICAPS (a) and assimilated simulated (b) 850 hPa wind fields (unit: m·s$^{-1}$) during the simulation period and their correlation coefficients of u and v components ($r_u$、$r_v$) given in the upper right corner

**Table 1.** Model parameterization settings

| Process | Number | Name |
|---|---|---|
| Longwave radiation | 4 | RRTMG (Mlawer et al., 1997) |
| Shortwave radiation | 3 | CAM (Collins et al., 2004) |
| Surface layer | 1 | MM5 Monin-Obukhov (Pahlow et al., 2001) |
| Land surface | 2 | Unified Noah (Chen et al., 2010) |
| Boundary layer | 1 | YSU (Shin et al., 2012) |
| Chemistry and aerosols | 9 | CBMZ and 4-bin MOSAIC (Sha et al., 2022) |
| Photolysis | 2 | Fast-J (Wild et al., 2000) |
| Sea salt emission | 2 | MOSAIC/SORGAM (Fuentes et al., 2011) |
| Dust emission | 13 | GOCART (Zhao et al., 2010) |
| Biogenic emission | 3 | MEGAN  (Guenther et al., 2006) |

**2.2 Four-dimensional data assimilation**

The accuracy of the meteorological field is crucial to reproduce realistic aerosol-cloud interaction scenarios, and thus a four-dimensional data assimilation approach is used to reduce the error of the simulated meteorological field. This approach utilizes relaxation terms based on the model error at observational stations to make the simulated meteorological fields closer to reality



(Liu et al., 2005), thus exerting positive effects on the simulation of atmospheric physical and chemical processes (Rogers et al., 2013; Li et al., 2016; Ngan and Stein, 2017; Zhao et al., 2020; Hu et al., 2022). The data used for assimilation are obtained from the NCEP operational global surface (https://rda.ucar.edu/datasets/ds461-0, last access: 19 March 2023) and upper-air

(https://rda.ucar.edu/datasets/ds351-0, last access: 19 March 2023) observation subsets, which contain meteorological elements such as altitude, wind direction, wind speed, air pressure, temperature, dew point and relative humidity.

### 2.3 Observational data

We use multiple observations to assess the impact of the four-dimensional assimilation and the ability of the model to reproduce meteorological fields, aerosol and cloud parameters. Precipitation data is taken from the Global Precipitation Measurement

(GPM) Level 3 dataset (https://disc.gsfc.nasa.gov/datasets/GPM_3IMERGDF_06/summary?keywords=Precipitation, last access: 30 May 2023), of which the daily accumulated high quality precipitation product is used in this study. Other meteorological elements are obtained from the Meteorological Information Combine Analysis and Process System (MICAPS) developed by the National Meteorological Center (NMC) of China (http://www.nmc.cn, last access: 19 March 2023), with 12 h temporal resolution and 11 vertical layers, containing meteorological elements such as wind field, height, temperature and

temperature dew point difference. Near-surface $PM_{2.5}$ data are obtained from the National Urban Air Quality Real-time Release Platform of China National Environmental Monitoring Centre with 1 h temporal resolution (https://air.cnemc.cn:18007, last access: 19 March 2023). The cloud parameters are obtained from the MODIS Level-2 Cloud (MOD06_L2) product (https://ladsweb.modaps.eosdis.nasa.gov/search/order/1/MOD06_L2--61, last access: 19 March 2023), from which we select cloud droplet effective radius (CER), cloud optical thickness (COT), CLWP and cloud phase data at 1 km resolution, as well

as cloud top height (CTH), cloud top temperature (CTT) and cloud top pressure (CTP) at 5 km resolution. The CER, COT and CLWP are retrieved from 2.1 μm wavelength, which is the default value in the product (1.6 μm and 3.7 μm wavelength retrievals are also available).

Spatial correlation analysis (Pearson product-moment coefficient), Pearson linear correlation analysis, and root mean square error (RMSE) are used to assess the spatial and temporal correlations of the simulated and observed values as well as

the error of the simulated values relative to the observations. To calculate these parameters, it is necessary to unify the spatio-temporal coordinates of the simulated and observed data. We interpolate the observations into the same horizontal grid as the simulated data and interpolate the simulated data into the same vertical pressure layer as MICAPS when evaluating the meteorological field. For the MODIS data, we select the reliable cloud retrievals according to the approach of Saponaro et al. (2017): (1) selecting only liquid-phase cloud parameters and (2) filtering out transparent-cloudy pixels (COT < 5) to limit

uncertainties (Zhang et al., 2012). The same filtering also applied to model outputs when doing evaluation against MODIS data. Since MODIS provides no information of $N_d$, we refer to the approach of Brenguier et al. (2000) and Quaas et al. (2006) utilizing MODIS COT and CER to calculate (when comparing the MODIS data, we did the same for the simulated values to obtain the simulated $N_d$, so that the two can be compared directly):

$$N_d = \gamma \cdot COT^{0.5} \cdot CER^{-2.5} \tag{1}$$



where $\gamma$ is an empirical constant with the value of $1.37 \times 10^{-5}$ m$^{-5}$. Moreover, due to the discontinuity of MODIS data, we matched the simulated data with MODIS data in spatio-temporal coordinates for evaluation (i.e., the simulated value is valid only when the MODIS data is valid in that spatio-temporal coordinate, otherwise the simulated value is set as the missing and does not participate in the calculation). Due to the differences in satellite retrievals and model parameterization, the simulated liquid-phase clouds are often defined based on certain thresholds when comparing with satellite-retrieved data, e.g. Roh et al.

(2020) classified the clouds with CLWC > 1 mg m$^{-3}$ and cloud ice water content (CIWC) < 1 mg m$^{-3}$ as liquid-phase clouds. In this study, based on the selection of column COT $\geqslant$ 5 that matched with MODIS filtering, the vertical layers (48 layers in total) with cloud optical thickness for water (COTW) > 0.1 and cloud optical thickness for ice (COTI) < 0.01 at each grid point and each time are selected as liquid-phase cloud layers, and the highest layer meeting this condition is defined as the simulated cloud top (this filtering is only used for comparison with MODIS data, and the analysis of aerosol-cloud interactions in liquid-

phase clouds in this study is strictly limited to CLWC > 0 and CIWC = 0).

## 3 Results and Discussion

### 3.1 Evaluation of assimilation effect and simulation result

Due to limitations in the resolution of observational data (e.g., MICAPS gridded upper-air meteorological field data with a resolution of 2.5°) and data availability (e.g., only terrestrial near-surface observations of PM$_{2.5}$ are available), we utilized

outer domain simulations when evaluating the model results. For aerosol-cloud analysis in Section 3.3 and beyond, we employed finer inner domain simulations.

    Four-dimensional data assimilation directly alters the meteorological field simulation and thereby affects the aerosol and cloud simulation. We first examine the effect of assimilation to clarify whether assimilation brings more confidence to the study. Figure 2 presents the vertical distribution of the simulated and observed meteorological elements before and after

assimilation, as well as the RMSE of the simulated relative MICAPS observations at each layer. The four-dimensional assimilation exerts slight effect on the height field (Fig. 2a), which is almost absent at the lower layers and exhibits slight increase at the upper layers in the RMSEs with respect to the observations compared to the unassimilated. In contrast to the height field, the assimilation presents significant improvements to the simulated temperature (Fig. 2b), and the RMSEs are effectively reduced by assimilation at all layers. The effects of assimilation on the temperature dew point difference (Fig. 2c)

and wind v component (Fig. 2e) exhibit reduced low and high layer RMSEs and enlarged middle layer RMSEs, while the effects on wind u component (Fig. 2d) exhibit reduced low and middle layer RMSEs and enlarged high layer RMSEs. As the complexity of atmospheric physical and chemical processes and data errors resulted from processes such as observation and station data gridding, the assimilation effects revealed by the evaluation are not uniformly positive, but overall exhibit positive effects. These positive effects are evident below 800 hPa, which especially helps to capture low clouds that dominate in winter.




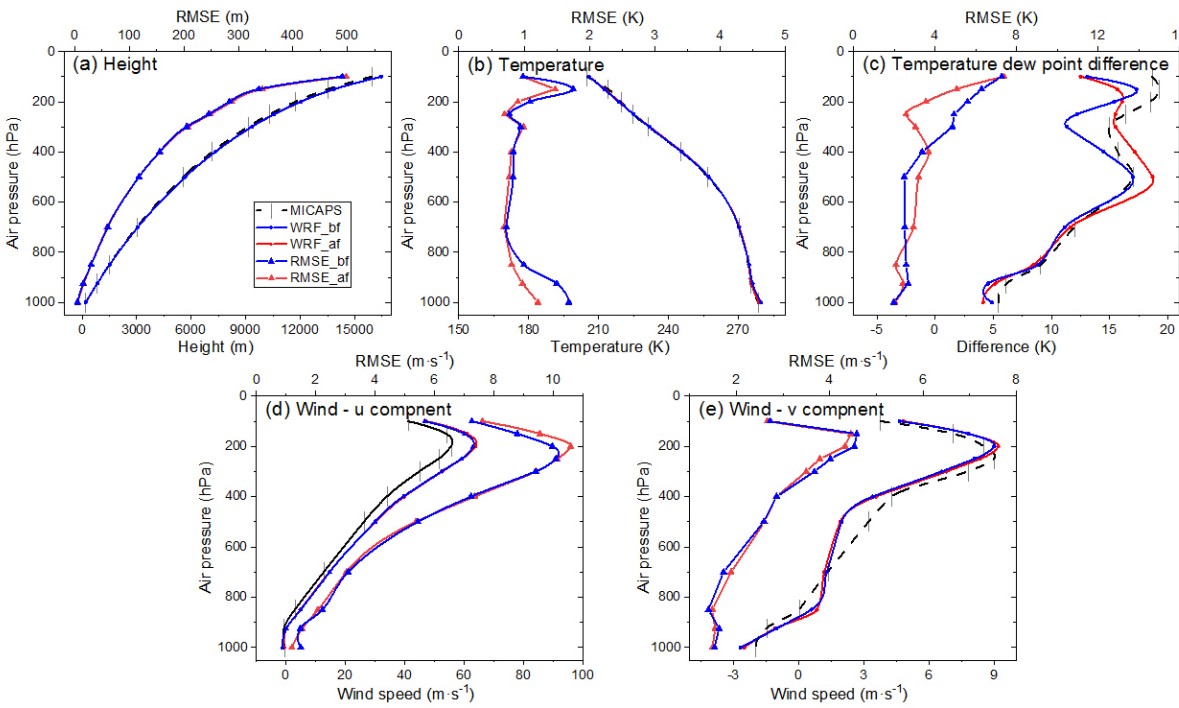

**Figure 2.** MICAPS and simulated height (a), temperature (b) and temperature dew point difference (c) as well as u (d) and v (e) components of wind and their RMSE vertical distribution before (blue line) and after (red line) assimilation

As shown in Fig. 3, assimilation exerts positive influences on precipitation, aerosol emission (mainly natural aerosols such as dust and sea salt), transport, and deposition. The RMSE of simulated and observed precipitation (Fig. 3a-c) is reduced by 60.4% after assimilation. As reported by previous studies, the WRF-Chem simulated AOD (Fig. 3e-f) is underestimated compared to MODIS retrievals (Fig. 3d) due to the differences in obtaining methods and calculations, and this underestimation is clear in the low-value region (Kumar et al., 2014; Soni et al., 2018; Krishna et al., 2019). Overall, the model reasonably reproduces the MODIS AOD distribution, and after assimilation, the simulation matches better with MODIS retrievals, especially in EC and ECO regions near 30°N that are of concern in this study. The simulated and observed near-surface $PM_{2.5}$ distributions presented in Fig. 3g-i indicate that the simulated $PM_{2.5}$ before and after assimilation both reasonably reproduce the observed patterns, but the simulation without assimilation underestimates the $PM_{2.5}$ concentration. Supported by assimilation, the model better reproduces the meteorological field as well as atmospheric physical and chemical processes, thus effectively optimizing the aerosol simulation, with the RMSE between the simulated and observed near-surface $PM_{2.5}$ reduced from 35.22 $\mu g\ m^{-3}$ to 27.73 $\mu g\ m^{-3}$. To evaluate the effect of assimilation on the simulation of $PM_{2.5}$ temporal variation, 16 stations with relatively continuous observation (Fig. 3h) are selected evenly from the model domain (Fig. 4). In general, the simulations before and after assimilation both reasonably reproduce the temporal variation of near-surface $PM_{2.5}$, and the correlation between simulated and observed $PM_{2.5}$ at all stations pass the test at 99% significance, whereas the simulations



before assimilation overall underestimate the PM$_{2.5}$ concentration. With assimilation, the simulated PM$_{2.5}$ concentrations are generally closer to the observations, and the correlation coefficients between the simulated and the observed have increased in 13 of the 16 stations, while the average correlation coefficient of the 16 stations has increased from 0.48 to 0.58.

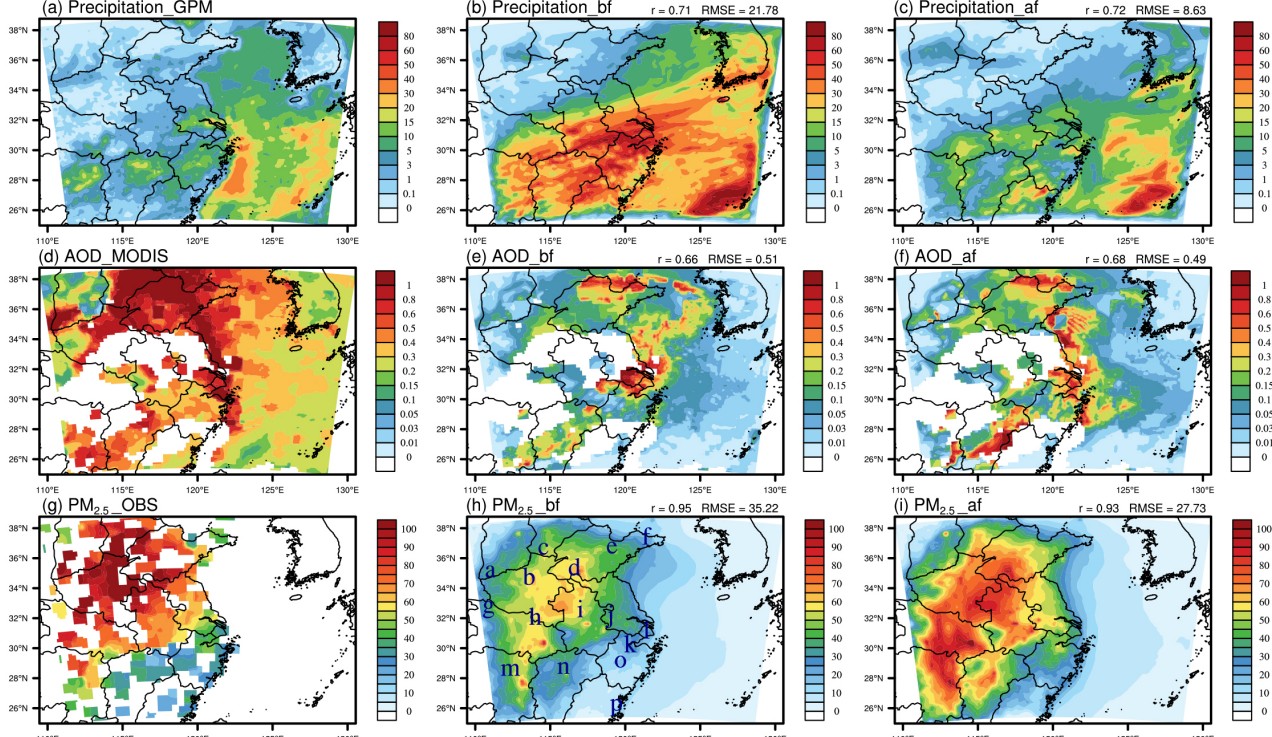

**Figure 3.** Distributions of accumulated precipitation (unit: mm, a-c), average AOD (dimensionless, d-f) and near-surface PM$_{2.5}$ (unit: μg·m$^{-3}$, g-i) during the simulation period from the observation (a, d and g) as well as before (b, e and h) and after (c, f and i) assimilation of the meteorological fields (r and RMSE at the up-right corner represent the spatial correlation coefficient and root mean square error of the observed and the simulated, respectively, where RMSE is in the same unit as the variable in the figure. The markers a-p in Fig. 3h represent the locations of the stations in Fig. 4)





**Figure 4.** Temporal variations of near-surface $PM_{2.5}$ observed (black line) and simulated before (blue line) and after (red line) assimilation of meteorological fields, at each site. The r and p values represent the correlation and significance of the observation and simulation, respectively, and subscripts "bf" and "af" represent simulated before and after assimilation

Figure 5 illustrates a comparison of cloud parameters between simulated and MODIS data. The model reasonably reproduces the spatial pattern of CER and $N_d$, i.e. low over land and high over ocean. Compared to MODIS, the model overestimates oceanic CER and underestimates oceanic $N_d$. Additionally, the model overall reasonably reproduces the values and distribution of CLWP, COT and cloud top parameters.

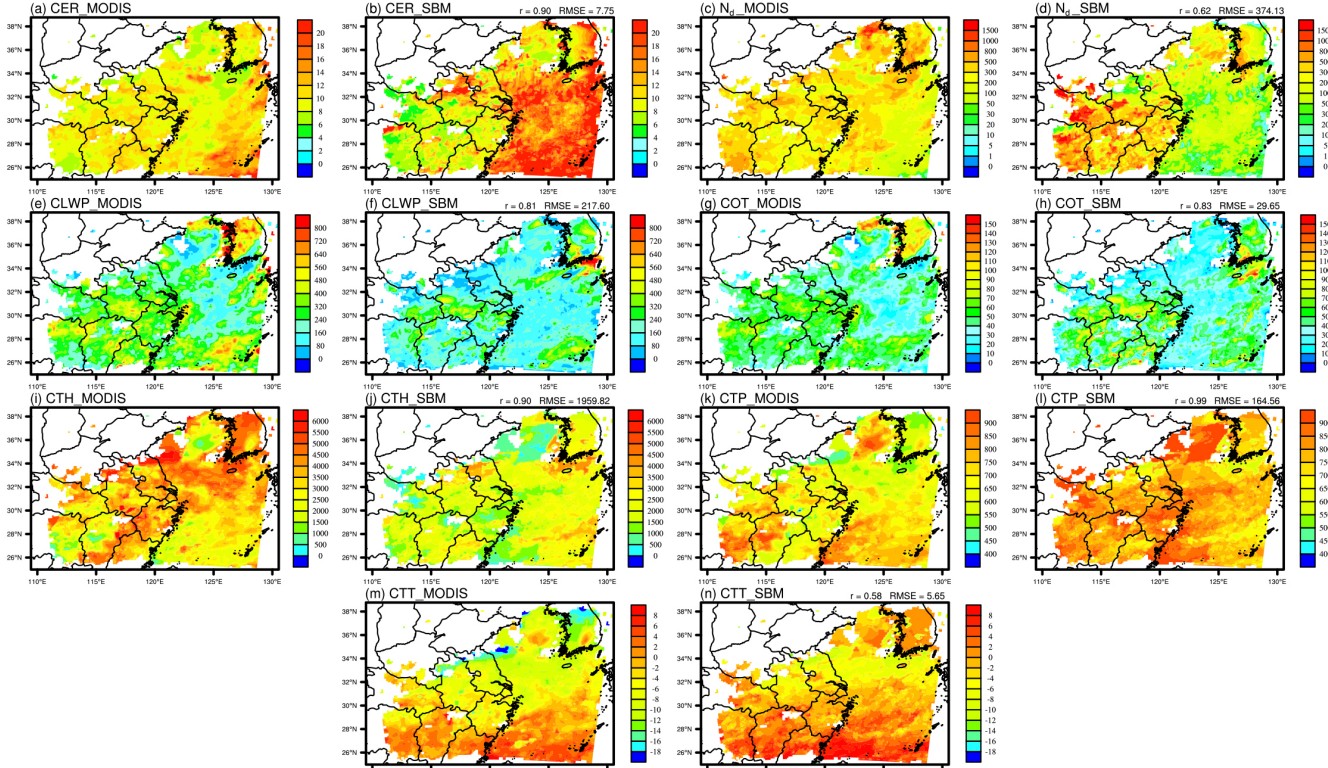

**Figure 5.** MODIS and simulated CER (a and b, in μm), $N_d$ (c and d, in cm$^{-3}$), CLWP (e and f, in g·m$^{-2}$), COT (g and h, dimensionless), CTH (i and j, in m), CTP (k and l, in hPa) and CTT (m and n, in °C) distributions (r and RMSE on top right represent the spatial correlation coefficient and root mean square error of the simulated and MODIS data, where RMSE is in the same unit as the variable in the figure)

Overall, the model reasonably reproduces the observed meteorological, aerosol and cloud information, providing a basis of confidence for aerosol-cloud interaction analysis based on the simulation results.

## 3.2 Aerosol and cloud droplet distribution in EC and ECO

The aerosol physical and chemical processes, aerosol-cloud interactions, and consequent aerosol and cloud droplet distribution characteristics in EC and ECO (Fig. 6) exhibit distinct discrepancies due to the differences in aerosol properties, topography, and meteorological fields. EC aerosols are mainly primary and secondary aerosols produced by anthropogenic emissions (Fig. 7 a-d and i-l), with small initial particle size. Under the conditions of multiple atmospheric supersaturation pathways, these small particles can be activated into cloud droplets, but the limited water vapor hinder further growth of cloud droplets. ECO aerosols are mainly transported from EC (Fig. 7 e-g and n-p), implying that most aerosols are anthropogenic aerosols with small particles. Meanwhile, compared to EC, there exist a higher proportion of large particles in ECO due to sea salt emissions



(Fig. 7 h and m). In addition, the abundant water vapor in ECO provides favorable conditions for aerosol activation and cloud

droplet growth, with much more cloud droplets above 12.7 μm radius than that in EC though the total cloud droplet is much

lower.

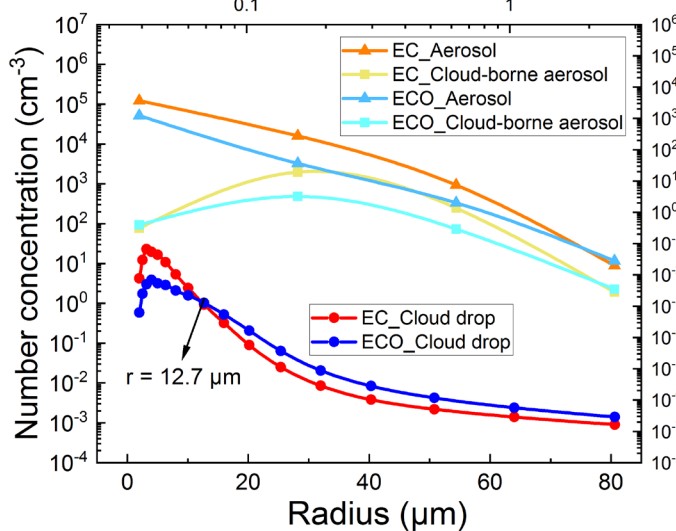

**Figure 6.** Size distributions of cloud droplets, total aerosols and activated (cloud-borne) aerosols in EC and ECO (bottom

and left axes correspond to cloud droplets, right and upper axes correspond to aerosols)






**Figure 7.** EC (a-d and i-l) and ECO (e-h and m-p) sulfate, nitrate, ammonium, chloride, sodium, OC, BC and other inorganic (dust) aerosols (in μg·m$^{-3}$) distributions


EC and ECO also exhibit clear differences in the spatial distribution of aerosols and cloud droplets. EC aerosol mainly originates from surface emissions, so its aerosol number concentration ($N_{aero}$) gradually decreases from surface to upper layer (Fig. 8a), while ECO aerosols are mainly transported from EC, so $N_{aero}$ hotspot in ECO is located at the transport altitude near 1800 m above sea level (Fig. 8b). In addition to aerosol number and size, atmospheric supersaturation is a determinant of

aerosol activation. EC and ECO show clear differences in the atmospheric supersaturation pathway. In EC, the main pathways of supersaturation include (1) atmospheric convection, which acts mainly in the areas with relatively strong updrafts and high water vapor content below 4000m altitude. Above 4000 m, the lack of water vapor makes it difficult to supersaturate even



with strong updrafts (Fig. 8e, i and k). (2) Water vapor and temperature changes caused by advection, which mainly works in the region of high water vapor content at tens of meters to 1000 m above surface. (3) Long-wave radiative cooling at surface,

which acts mainly at night or early morning (Fig. S1), leads to a high supersaturation of the atmosphere (the disappearance of this effect during the daytime makes the temporal average supersaturation near surface relatively low). The high aerosol concentration and supersaturation makes the high $N_d$ near surface (Fig. 8c). (4) Topographic uplift, the forced uplift of topography makes the atmosphere more susceptible to reach supersaturation. The main supersaturation pathways in ECO are convection and advection. Due to the abundant water vapor content, even though the ECO vertical convection is weak, the

relatively strong updraft area near 28°N at 2000-4000 m elevation generates much higher supersaturation than the EC (Fig. 8e-f and i-l), however, no $N_d$ hotspot is found in this area due to low aerosol concentration (Fig. 8b and d). In winter, at least during this study period, the effect of atmospheric convection on ECO aerosol activation is weak, and temperature and water vapor changes due to atmospheric advection are the dominant factors for ECO aerosol activation during this period.

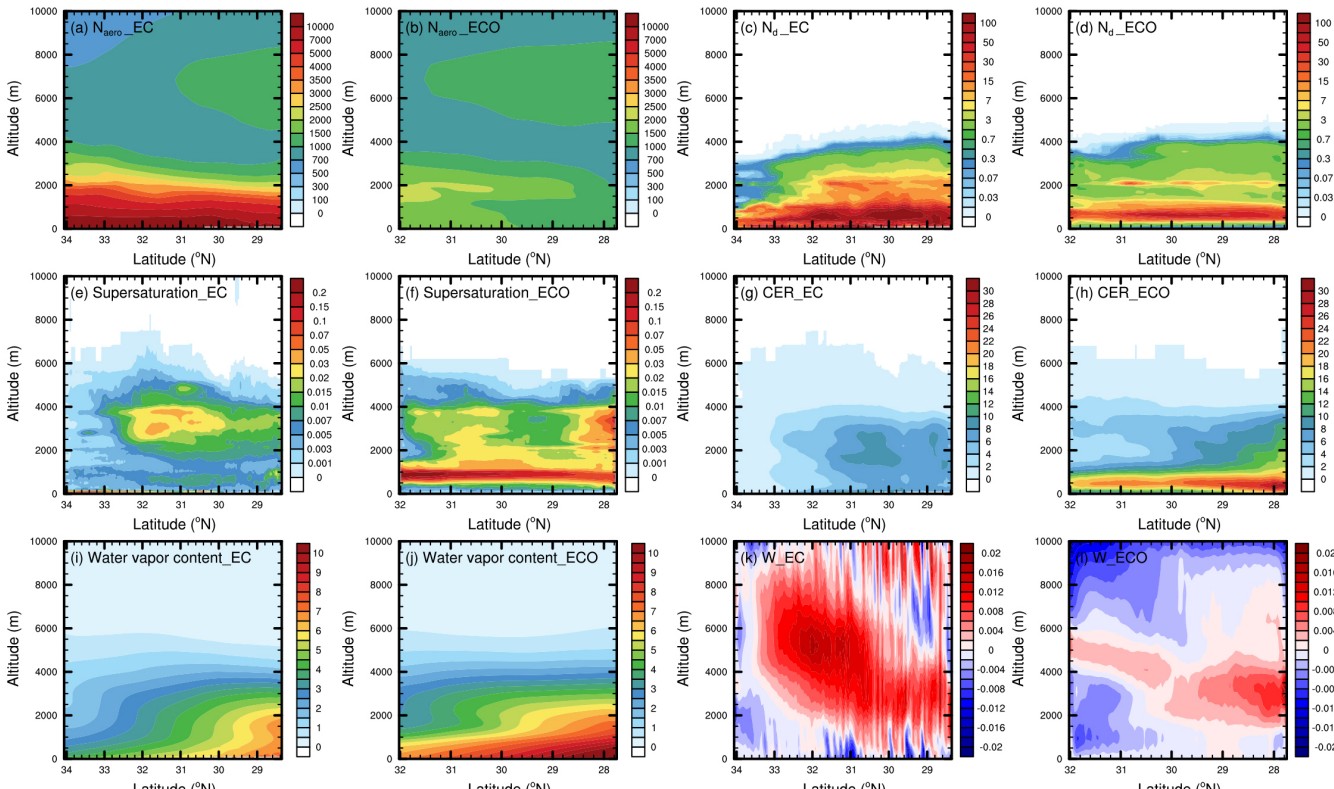

**Figure 8.** EC and ECO aerosol number concentration (in cm$^{-3}$, a-b), $N_d$ in liquid-phase cloud (in cm$^{-3}$, c-d), atmospheric supersaturation (in %, e-f), CER (in μm, g-h), water vapor content (in g·m$^{-3}$, i-j) and vertical wind speed (in m·s$^{-1}$, k-l) distributions



### 3.3 Aerosol activation of liquid-phase clouds in EC and ECO

To explore the responses of EC and ECO clouds to aerosols and their influencing factors, we perform a statistical analysis on aerosols, clouds and meteorological elements for the grid points with liquid-phase cloud at each time. The statistics are based on each vertical layer and the column, respectively, with the former providing abundant samples that are more favorable to avoid fluctuations from individual extreme processes and to obtain the average state of the liquid-phase clouds, and the latter facilitating direct comparisons with information such as satellite retrievals.

Aerosol activation is the first step of aerosol-cloud interaction, and we analyze the variation of $N_d$ with aerosol based on the statistics for each vertical layer (the first two lines of Fig. 9) and for the column (the third line of Fig. 9), respectively. At low $N_{aero}$, aerosols promote cloud droplet increase by acting as CCN. As aerosols and cloud droplets increase, more small aerosols heightens the requirement of atmospheric supersaturation for aerosol activation, and the consumption of water vapor from cloud droplet growth makes it more difficult for the atmosphere to reach supersaturation, thus suppressing aerosol activation. As shown in Fig. 8a and e, $N_d$ in both EC and ECO exhibit the general trend of increasing first and then decreasing with increasing $N_{aero}$, but there are some differences between EC and ECO. In EC (Fig. 8a-d), multiple atmospheric supersaturation pathways and abundant aerosols allow $N_d$ to maintain a more persistent increasing trend with $N_{aero}$ until average $N_d$ reaches 6000 cm$^{-3}$. In addition, aerosol activation is not completely suppressed in the near-surface with high aerosol concentrations ($N_{aero}$ above 40,000 cm$^{-3}$), and aerosols can still be activated at  high supersaturation caused by the effects of longwave radiative cooling (the diurnal variation of this effect is also one of the main reasons for the fluctuation of $N_d$ with $N_{aero}$) and terrain uplift (the high topographic gradient areas where this action takes effect are also typically characterized by high concentrations of aerosol accumulation), etc. In ECO (Fig. 9e-h), the dominance of water vapor variation on aerosol activation is more pronounced, and the supersaturation shows a steady decreasing trend with increasing $N_{aero}$. Once $N_{aero}$ exceeds 10000 cm$^{-3}$ (average $N_d$ reaches 3000 cm$^{-3}$), the increase in small aerosols and the decrease in supersaturation prevent its $N_d$ from continuing to increase and $N_d$ starts to show a decreasing trend. The statistics on the column (Fig. 9i-l) also show that the variation of $N_d$ with AOD in EC is much flatter, while $N_d$ decreases rapidly after AOD exceeds 0.8 in ECO. It is worth mentioning that the high AOD value in EC is dominated by large particles, which is one of the reasons why the trend of EC $N_d$ decreasing with AOD is flatter, whereas the high AOD value in ECO depends largely on the transport of EC aerosols, and the small particles that are more easily transported make its high AOD value mostly attributed to a large amount of small particles, which exacerbates the decrease ECO $N_d$ with AOD.



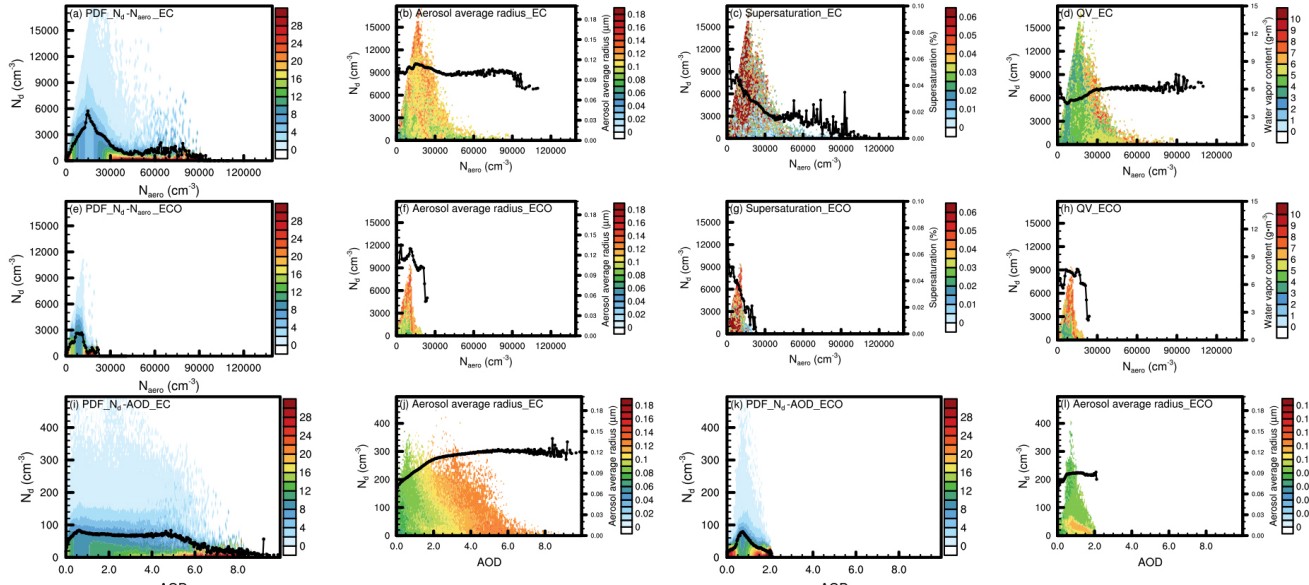

**Figure 9.** Probability density distribution functions (sum of probabilities corresponding to 1 for each $N_{aero}$ or AOD value) and means (lines in a, e, i and k) of the simulated $N_d$ relative to $N_{aero}$ (a and e) and AOD (i and k), as well the values (corresponding to the left and bottom axis) of aerosol average radius (b, f, j and l), supersaturation (c and g) and water vapour content (d and h) along with their mean changes (lines in figures, corresponding to the right and bottom axis) relative to $N_{aero}$ (the first two lines) and AOD (the third line) in EC (a-d and i-j) and ECO (e-h and k-l). The first two lines are based on statistics for each vertical layer, and the third line is based on statistics for column

To investigate the influence of meteorological conditions on aerosol activation, a statistically analysis on the variation of $N_d$ and $N_{aero}$ ratio with $N_{aero}$ for different zonal wind speed (U), meridional wind speed (V), vertical wind speed (W), temperature, water vapor content, as well as temperature and water vapor changes at that time compared to the last time are presented in Fig. 10. In EC, when both zonal and meridional wind speeds are close to 0 m·s$^{-1}$, a large number of aerosols accumulate and strong aerosol activation can occur under the effect of radiative cooling at surface. Exceptionally, strong aerosol activation also occurs when the zonal wind speed < -6 m·s$^{-1}$ or meridional wind speed < -8 m·s$^{-1}$, the former due to large amounts of water vapor from the ocean brought by easterly winds (Fig. S2b) and the latter due to cold air brought by northerly winds (Fig. S2a) and uplift caused by the south high and north low topography in EC (Fig. 1). The overall strong aerosol activation is exhibited at higher vertical wind speeds, but aerosol activation is also found when the vertical airflow is weak or dominated by downdraft due to the influence of advection at tens of meters to 1000 m above surface, topographic uplift, and long-wave radiative cooling at surface. Aerosol activation is hardly occur in the high-altitude downdraft dominated areas where $N_{aero}$ < 10000 cm$^{-3}$ and W < 0 m·s$^{-1}$. Liquid-phase clouds of EC appear mainly under the conditions of -5-14 °C temperature and 3-13 g·m$^{-3}$ water vapor content. Ice phase processes occur frequently below -5 °C, and in addition the



atmosphere is difficult to reach supersaturation in areas with water vapor content below 3 g·m⁻³ or water vapor content above 13 g·m⁻³ and temperature above 14 °C (temperature and water vapor content show generally consistent spatial distribution). The increase in water vapor and decrease in temperature contribute to EC aerosol activation, but the diversity of atmospheric supersaturation pathways enables aerosol activation to occur even at reduced water vapor or elevated temperatures.

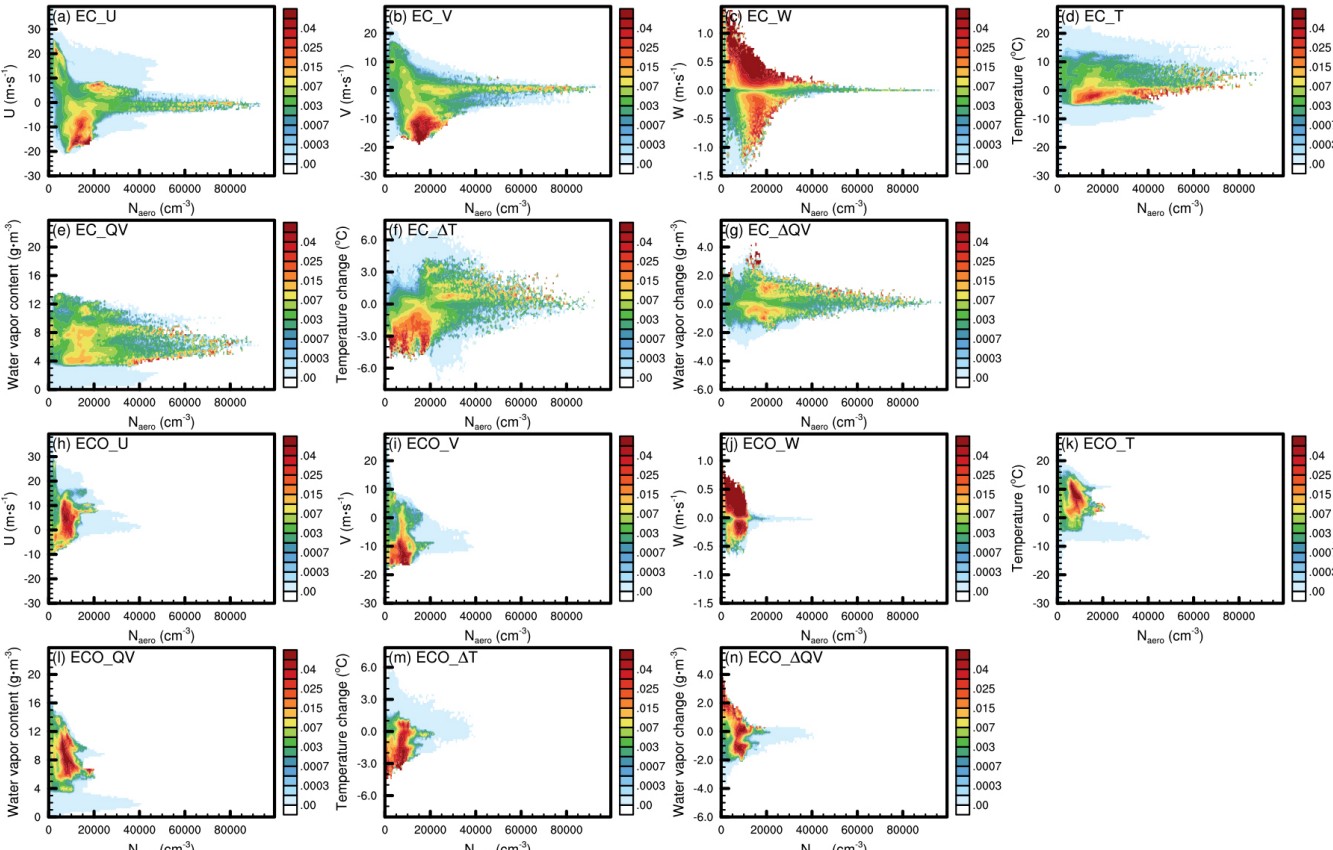

**Figure 10.** Variation of EC (a-g) and ECO (h-n) $N_d$ to $N_{aero}$ ratio (unit: cm⁻³·cm³) with $N_{aero}$ at different U-wind (a and h), V-wind (b and i), W-wind (c and j), temperature (d and k), water vapor content (e and l), temperature variation (f and m) and water vapor variation (g and n)

In ECO, the zonal wind speed favourable to aerosol activation is below 0 m·s⁻¹ or 0-12 m·s⁻¹, with the former ensuring the supply of water vapor and the latter providing more abundant aerosols, while at zonal wind speed above 12 m·s⁻¹, the excessively dry air from land makes the atmosphere difficult to reach supersaturation, despite the large amount of aerosols brought by the westerly wind. The meridional wind speed suitable for ECO activation is mainly below -8 m·s⁻¹, the cold air brought by strong northerly winds makes it easier for the atmosphere to reach supersaturation. The abundance of water vapor makes ECO more susceptible to reach supersaturation by updrafts, making its activation exhibit a high sensitivity to vertical wind speeds. The temperature and water vapor conditions suitable for liquid-phase clouds in ECO are the same as in EC, with



the difference that the abundant water vapor in ECO results in clearly stronger activation at higher temperatures than in EC.

### 3.4 Impact of aerosols on development of liquid-phase clouds

Aerosol activation alters cloud droplet size distribution and consequent changes in cloud microphysical and dynamical
processes, which is also known as rapid adjustment (Heyn et al., 2017; Mulmenstadt and Feingold, 2018). We discuss the
variations of CLWC and CER with increasing $N_d$ for precipitation clouds (raindrop number concentration above 0 m$^{-3}$) and
non-precipitation clouds (raindrop number concentration of 0 m$^{-3}$). In EC, the multiple atmospheric supersaturation pathways
and abundant aerosols cause wide $N_d$ range (Fig. 11a-d). Higher $N_d$ allows the precipitation cloud to reach a higher CLWC in
EC than that in ECO, but the influence of precipitation and evaporation gradually slows down the increase of CLWC with $N_d$.
When $N_d$ exceeds 16000 cm$^{-3}$, CLWC decreases rapidly as a result of the precipitation and higher evaporation triggered by the
small cloud droplets. Compared to precipitation clouds, non-precipitation clouds in EC have lower CLWC, but the absence of
precipitation process and weaker evaporation due to the lower difference in intracloud and ambient water vapor pressure result
in a more steady increase in their CLWC with $N_d$, and after $N_d$ exceeds 12000 cm$^{-3}$, the decreasing tendency of CLWC is less
pronounced. In ECO (Fig. 11e-h), although the supersaturation pathway limits the increase of $N_d$ and CLWC, abundant water
vapor causes its CLWC to increase faster at low $N_d$ as well as a weaker decrease in CLWC at high $N_d$. In addition, the scarcity
of non-precipitating cloud samples under abundant water vapor results in a stronger volatility in CLWC.

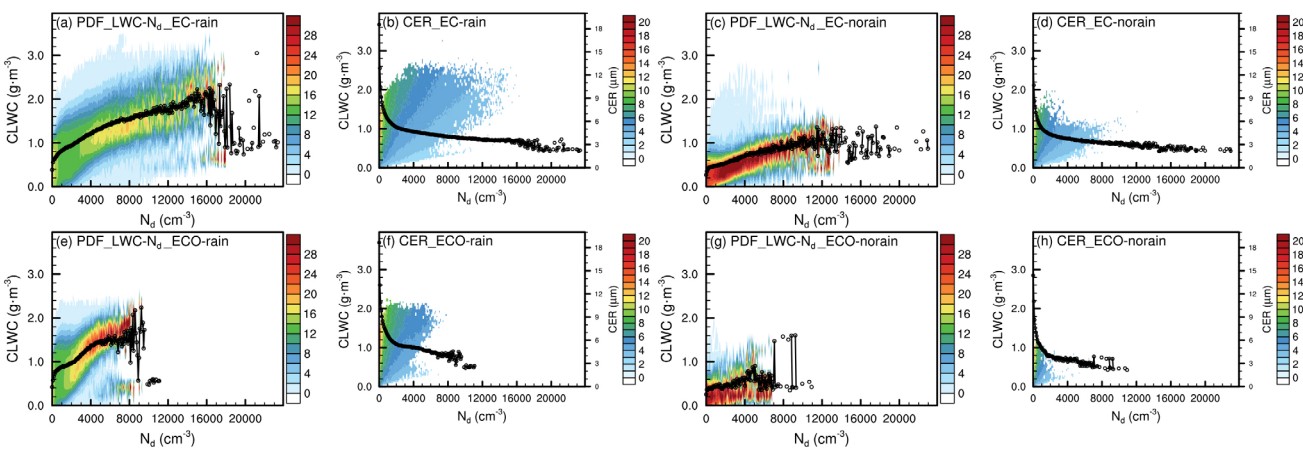

**Figure 11.** Probability density distribution functions (sum of probabilities corresponding to 1 for each $N_d$ value) and means
(lines in the figures) of CLWC relative to $N_d$ (a, c, e, and g), as well as the CER values (b, d, f, and h, corresponding to the left
and bottom axis) and their mean changes relative to $N_d$ (lines in the figures, corresponding to the right and bottom axis) of
precipitation clouds (a-b and e-f) and non-precipitation clouds (c-d and g-h) in EC (a-d) and ECO (e-h)

We further examine the variations of CLWP and its related elements with AOD based on the statistics of the column , in
order to provide direct comparisons with satellite retrievals. In EC, heavy precipitation (Fig. 12d) causes a rapid decrease in




CLWP at low AOD (Fig. 12a). Once AOD exceeds 2, small CER inhibits precipitation and the decrease in CLWP, sufficient

aerosols and multiple atmospheric supersaturation pathways lead to the stabilization of CLWP. With increasing AOD, CER and $N_d$ in EC exhibit decreasing followed by increasing and increasing followed by decreasing trends, respectively, with both exhibiting significant fluctuations at high AOD values (since Fig. 12c samples $N_d$ based on different intervals of ADO and CLWP, its variation of $N_d$ with AOD is close but not identical to Fig. 9i). When AOD is lower than 4, the variation of rain water path (RWP, i.e., column rain water content) is dominated by the variation of $N_d$, i.e., the increase of $N_d$ leads to the

decrease of CER and RWP. The dominance of large aerosol particles at high AOD (fig 9j) gradually increases the RWP after AOD exceeds 4, but as precipitation consumes the CLWP, the RWP gradually decreases after AOD exceeds 8. Compared to EC, ECO, where the supersaturation pathway is more monotonous, is also more monotonous in the variation of cloud parameters, i.e., both CLWP and RWP show a steady decreasing trend with increasing AOD. Due to the characteristics of the model calculation and the limitation of the column statistics, the obtained samples of non-precipitation clouds (RWP = 0 g·m$^{-}$

$^2$) are rare, and the CLWP of all clouds (precipitation and non-precipitation clouds, Fig. 12l and p) shows almost the same trend as that of precipitation clouds. In these samples, EC non-precipitation clouds exhibit cloud parameter variations that are generally similar to precipitation clouds but more moderate, mainly due to weaker in-cloud processes. ECO, due to a limited number of samples, displays more pronounced variations in cloud parameters for non-precipitation clouds compared to precipitation clouds.

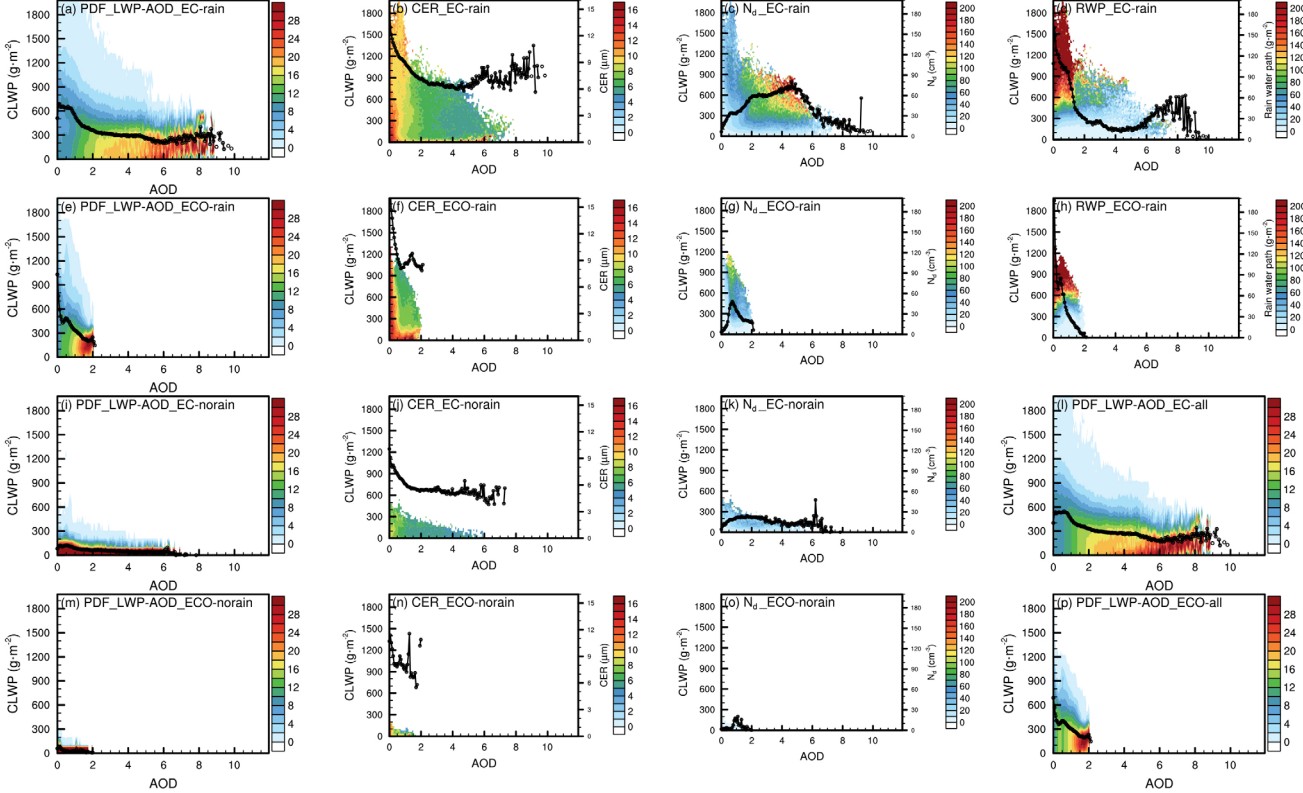





**Figure 12.** Probability density distribution functions (sum of probabilities corresponding to 1 for each AOD value) and means (lines in the figures) of CLWP relative to AOD (a, e, i, l, m and p), as well as the variation of CER (b, f, j and n), $N_d$ (c, g, k and o) and column rain water path (RWP, d and h) with AOD of precipitation clouds (a-h), non-precipitation clouds (i-k and m-o) and all clouds (l and p) in EC (a-d and i-l) and ECO (e-h and m-p)

Fig. 13 exhibits the effects of different meteorological and aerosol conditions on CLWC. The CLWC and $N_{aero}$ ratio under different meteorological and $N_{aero}$ conditions shows generally similar variation to the $N_d$ and $N_{aero}$ ratio in Fig. 10, with only some minor differences. Compared to the high activation ratios exhibited in Fig. 10, which tend to occur at medium $N_{aero}$, the high CLWC and $N_{aero}$ ratios at low $N_{aero}$ are more heavily weighted due to the more abundant water vapor supply and weaker evaporation when there are fewer but larger cloud droplets.

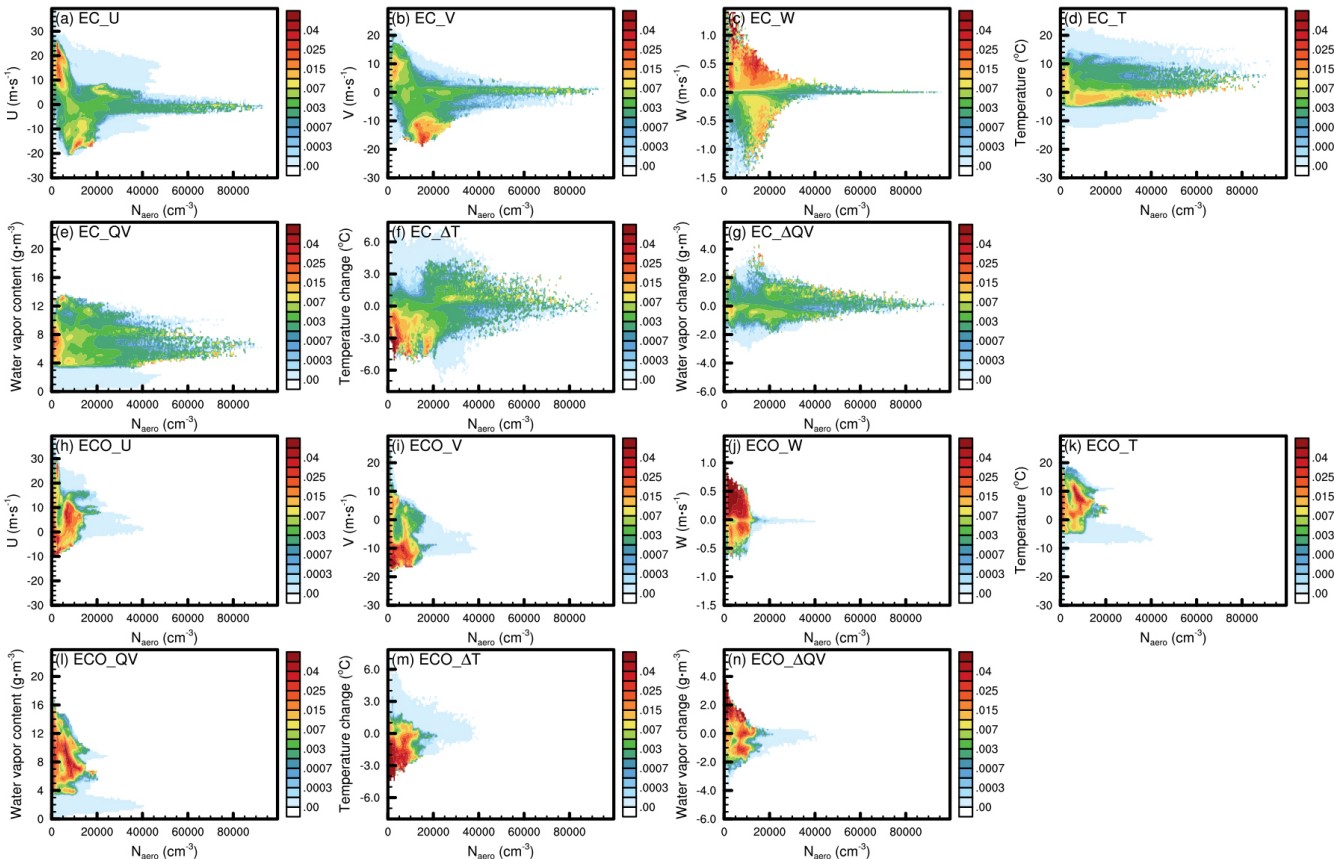

**Figure 13.** Variation of EC (a-g) and ECO (h-n) CLWC to $N_{aero}$ ratio (unit: mg·m$^{-3}$·cm$^3$) with $N_{aero}$ at different U-wind (a and h), V-wind (b and i), W-wind (c and j), temperature (d and k), water vapor content (e and l), temperature variation (f and m) and water vapor variation (g and n)



## 4 Conclusion

In this study, aerosol-cloud interactions in liquid-phase clouds over eastern China (EC) and its adjacent ocean region (ECO)
in winter are explored based on the WRF-Chem-SBM model in which a spectral-bin cloud microphysics (SBM) and online
aerosol module (MOSAIC) are coupled.

The impact of four-dimensional data assimilation on the simulation and performance of the coupling system is firstly
evaluated using multiple observations. Overall we see positive effects on the simulated meteorology by using data assimilation,
especially below 800 hPa, which is very beneficial for the simulation of low clouds that dominate in winter. The simulations
of precipitation and aerosol are effectively improved by optimizing the meteorological field and the observed and simulated
precipitation and near-surface $PM_{2.5}$ RMSEs are reduced by 60.4% and 21.3%, respectively. By using four-dimensional
assimilation, the WRF-Chem-SBM model reasonably reproduces satellite-retrieved cloud parameters, and providing a
confidence basis for the analysis of aerosol-cloud interactions in this study.

Differences in meteorological, topographic, and aerosol conditions make EC and ECO aerosol activation distinctly
different. There are four main atmospheric supersaturation pathways in EC, namely updrafts mainly acting below 4000m
altitude, advection mainly operating at tens to 1000 m, long-wave radiative cooling near surface at night to early morning, and
topographic uplift in areas of large topographic gradient. The multiple supersaturation pathways produce a large number of
cloud droplets in EC, but the limited water vapor over land constrains the growth of cloud droplets, causing them to exhibit a
large total quantity but generally small size. In contrast, only convection and advection act as the two main supersaturation
pathways in ECO. The abundant water vapor in ECO produce higher supersaturation than EC with weaker updraft near 28°N
at 2000-4000 m elevation where convection dominates atmospheric supersaturation, but no $N_d$ hotspot appears in this area due
to the limitation of $N_{aero}$. The area around 1000 m, where both water vapor and aerosols are relatively abundant, is the $N_d$
hotspot in ECO, and the abundant water vapor provides powerful support for the cloud droplet growth, so that the ECO cloud
droplets over 12.7 μm radius are much more than those in EC with much fewer total quantity than EC.
We explore the effects of different meteorological and aerosol conditions on cloud formation and development through
the statistics of liquid-phase cloud samples. Our analysis of aerosol activation shows that in both EC and ECO, $N_d$ exhibits a
trend of increasing and then decreasing with increasing $N_{aero}$. The difference between EC and ECO is that multiple
supersaturation pathways and abundant aerosols in EC cause $N_d$ to exhibit a much robust increasing trend compared to ECO
at low $N_{aero}$ and strong fluctuations at high $N_{aero}$. While in ECO, the strong dependence of aerosol activation on water vapor
content makes $N_d$ maintain only an unsteady increasing trend with $N_{aero}$. Once $N_{aero}$ exceeds 10000 $cm^{-3}$ (average $N_d$ exceeds
3000 $cm^{-3}$), the consumption of water vapor by cloud droplet growth makes it difficult for the atmosphere to reach
supersaturation, together with the higher percentage of small aerosols at high $N_{aero}$, ECO $N_d$ increase is gradually suppressed.
In EC, favourable meteorological conditions for aerosol activation include: (1) aerosol accumulation under static or weak wind
conditions and high supersaturation due to radiative cooling at surface, (2) moist air brought by strong easterly wind, (3)
cooling and topographic uplift due to strong northerly wind, and (4) strong updraft. In ECO, the meteorological conditions



suitable for aerosol activation include (1) aerosol-rich and not excessive dry airflow from moderate westerly wind, (2) cooling due to strong northerly wind, and (3) updraft. In general, the contribution of cooling to aerosol activation is more pronounced relative to humidification in EC and ECO.

We further discuss the effect of aerosol-induced $N_d$ variations on CLWC. For precipitation clouds, CLWC increases and then decreases with increasing $N_d$ in both EC and ECO. The high $N_d$ supported by multiple supersaturation pathways in EC causes higher CLWC than ECO at lower water vapor content, while ECO exhibits a more rapid increase in CLWC before the increase in CLWC stagnates.  Due to the weaker intracloud processes, the CLWC variation trend of EC non-precipitation clouds is gentler than that of precipitation clouds. In contrast, ECO shows more pronounced fluctuations in its non-precipitation cloud CLWC variation than precipitation clouds due to the scarcity of non-precipitation cloud samples under abundant water

vapor. We analyze the variation of CLWP with AOD based on the statistics on column. In both EC and ECO, the strongest precipitation occurs at AOD less than 1, causing CLWP to decrease rapidly with AOD. With increasing AOD, the increase in $N_d$ inhibits the reduction of CLWP in both EC and ECO, and this inhibition is more pronounced in EC where atmospheric supersaturation is less dependent on water vapor. As AOD increases further and $N_d$ begins to decrease, the lack of water vapor and the decrease in aerosol particle size cause CLWP in ECO to decrease rapidly, while the dominance of large particles at

high AOD in EC allows its CLWP to remain stable. Meteorological and aerosol conditions suitable for EC and ECO cloud development are basically the same as those suitable for aerosol activation, with the difference that high aerosol activation ratios occur mainly under moderate $N_{aero}$ conditions, whereas high CLWC and $N_{aero}$ ratios often occur under low $N_{aero}$ conditions in addition to moderate $N_{aero}$ conditions.

***Data availability.*** The model outputs are available upon request, the other data can be accessed from the websites listed in Sect. 2.

***Author contributions.*** JZ and XM designed and conducted the model experiments, analysed the result and wrote the paper. XM developed the project idea and supervised the project. XM, JQ and HJ proposed scientific suggestions and revised the

paper.

***Competing interests.*** One author (JQ) is a member of the editorial board of journal ACP, and the authors have no other competing interests to declare.

***Acknowledgements.*** This study is supported by the National Natural Science Foundation of China (Grants 42061134009 &41975002) and the Postgraduate Research and Practice Innovation Program of Jiangsu Province (Grant KYCX22_1151). The numerical calculations in this paper was conducted in the High-Performance Computing Center of Nanjing University of Information Science & Technology. We express our gratitude to Dr. Jiwen Fan of Argonne National Laboratory for providing the code for the WRF-Chem-SBM model and Prof. Qian Chen of Nanjing University of Information Science & Technology



for providing advice on this study. We are grateful to the National Aeronautics and Space Administration, the National Center
        for Environmental Prediction, MEIC Support Team, the Chinese National Meteorological Center and China National
        Environmental Monitoring Centre for providing the MODIS and GPM data, FNL and observation subsets, MEIC emission
        inventory, MICAPS data and PM$_{2.5}$ data respectively.

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
