# Peer review of "Exploring aerosol-cloud interactions in liquid-phase clouds over eastern China and its adjacent ocean using the WRF-Chem-SBM model"

_EGUsphere, 2023_

## Referee Comment (RC1)

**Review of "Exploring aerosol-cloud interactions in liquid-phase clouds over eastern China and its adjacent ocean using the WRF-Chem-SBM model" by Zhao et al., submitted to Atmospheric Chemistry and Physics (ACP)**

[Article#: acp-2023-2858]

This report contains general, major, and specific comments from this reviewer on the manuscript.

**A summary of the manuscript and general assessment:**

Recommendation: Reconsidered after major revisions

This study conducted WRF-Chem-SBM simulations to investigate aerosol-cloud interactions, particularly the relationship between aerosol number concentration and cloud droplet number concentration/liquid water content, focusing on the differences between eastern China (EC) and the adjacent ocean (ECO). The simulations reasonably reproduced the distributions of cloud properties and precipitation amounts obtained from satellite measurements, especially when a four-dimensional assimilation was applied. The authors identified dominant mechanisms for cloud development in the EC and ECO. The different relationships between aerosol number concentration and cloud droplet number concentration/liquid water content in EC and ECO are presented and the mechanisms for the differences are discussed.

This study is within the aims and scope of Atmospheric Chemistry and Physics (ACP), specifically the subject for "Aerosols, Cloud and Precipitation", the research activity for "Atmospheric modelling", the altitude range for "Troposphere", and the science focus on "both Chemistry and Physics".

Although I appreciate the authors' efforts in the model simulations using the state-of-the-art model for chemical-aerosol-cloud-precipitation interactions and the analysis, this manuscript has significant problems in readability and presentation, particularly using figures. It was very tough for me to read the whole manuscript and try to understand the contents in the presentation. I found that this manuscript was resubmitted to ACP after a rejection, and I reviewed the open discussion on the previous manuscript. I have no idea why the previous reviewers did not point out the presentation problems. However, I cannot ignore the problems because they greatly reduce the quality of the manuscript. Also, in my opinion, there are still major problems in the analysis approach of the simulation results and the discussion of the

significance of the results with those of other similar studies. More details are provided in the following paragraphs.

First, the presentation using figures has serious problems. The worst figure and caption are in Fig. 9, and I describe my complaints individually in the Major Comments below. The main problem with the figures is that there is little explanation of what is shown in most of the figures and how the values shown are calculated. As a result, almost every time I encountered figures while reading the manuscript, I had to speculate how the authors calculated and produced the plots. This was stressful and confusing. In most cases, I assumed simple spatial and/or temporal averages, not instantaneous values of the variables, because the date and time are not specified in the captions. However, there are some variables, such as positive supersaturation and cloud droplet effective radius (CER), that cannot simply be averaged over all grid points. For example, in Fig. 8, these variables are shown together with Naero and water vapor, which can be simply averaged over all grid points. I am very confused as to whether all variables were averaged over the same sampling or not, yet the authors use the figure to argue for their correlation. In particular, sampling supersaturation needs special care, because it could be negative even in the cloud. The values and the scientific implications of the averaged supersaturation depend heavily on how it is sampled for averaging.

I request the authors to revise the figures, their captions, and the text of the manuscript to explicitly state what is shown and how the values shown in all figures (including Supplementary Materials) were calculated to avoid confusion and misunderstanding by readers. This is one of the minimum requirements for scientific papers.

Second, the approaches to analyze the simulation results in Sections 3.3 and 3.4 have significant problems. It is difficult for me to tell all the problems, especially related to the statistical analysis, and how to improve them to the authors in these review comments, but I hope the manuscript will be improved as much as possible for reconsideration. First of all, the authors would perform statistical analysis to obtain the average states without fluctuations from individual extreme processes (line 318). However, Figs. 9-13 clearly show such fluctuations, probably due to the limited sampling volume for the conditions. The parts with fluctuations are reflected in the discussion and conclusions. If the authors want to exclude the effects of individual extreme processes, the conditions with low frequencies should be filtered out in the calculation of the statistics.

Related to the above problem, there is little information about the probability or frequency of the variables discussed in the sections, except for Nd and CLWP. This makes it difficult to understand how representative each point in the colored contours and lines in Figs.

9-13 is of the entire special and temporal sampling volume. For example, Fig. 10a shows a ratio of Nd to Naero averaged over small ranges (bins) of U and Naero, if my understanding is correct. However, this plot does not have the information of how frequent and representative each point is in the whole sample volume. For example, -20 m/s of U is probably a fairly rare case and not representative of what is simulated in EC. Additional levels of statistical analysis for frequency and sampling are needed to unravel the correlation with different weather conditions and finally understand the relationship between Naero and Nd-LWP.

There are some misleading interpretations of the statistical analysis. Please be careful when writing the explanation. For example, a higher mean does not always mean a higher frequency or probability.

Third, there is no discussion of what and how the use of the state-of-the-art model improved the simulations of aerosol-cloud interactions compared to other previous modeling studies as well as observations. This is necessary to explicitly demonstrate the scientific significance of this research. For example, as partially described in the introduction of the manuscript, conventional global aerosol transport models have had problems, e.g., they tend to overpredict the increase in LWP in response to increases in Na or Nd, compared to global-scale satellite observations (e.g., Quaas et al. 2009). Did the simulations in this study solve the problem or not? How similar or different is the obtained relationship between LWP and Na(AOD)orNd compared to those in high-resolution model simulations in other studies? In addition, there is no direct comparison of the relationship with satellite observations in this case, although the authors sometimes emphasize the advantages in the manuscript. Why was the relationship between LWP and Nd from the simulations not directly compared with that from Terra/Aqua MODIS satellite observations in this case?

**Other Major comments:**
1. Caveat when evaluating simulation performance based on RMSE and correlation to the coarser resolution gridded dataset (Fig. 2)

There is a caveat when evaluating the simulation performance based on the RMSE against the coarser resolution gridded data set than the model horizontal grid spacing in Fig. 2. The MICAP is horizontally gridded data with a resolution of 2.5 degrees. This basically means that the data does not contain information about variations at horizontal scales smaller than degrees, which can definitely occur in the real world. Thus, when MICAP is re-gridded into the model horizontal grid structure with finer grid spacing, the interpolated data cannot be fully used as the true value for evaluating spatial variations in the simulated fields containing finer-scale

variations represented in the model with finer grid spacing. Therefore, the calculated RMSE for Figure 2 has little validity, and the difference in RMSE calculated in such an approach does not guarantee worse or better performance of the simulations.

From this perspective, the RMSE and correlation against IMERG in Fig. 3 are okay, but those against PM2.5 have the same problem. In the case of PM2.5, the improvement of the mean bias by using the assimilation greatly reduces the RMSE, while it does not guarantee the improvement of the horizontal variations, especially at smaller scales, as shown by the similar correlation coefficients between the two simulation results.

2. Figure 9

At first glance, I could not understand what the color contours in these plots were showing at all because of the mismatch between the plots and the caption, as well as the poor wording in the caption. However, after struggling for several hours over days, I finally reached an understanding of what these plots would show to some degree, although I still do not have full confidence. First of all, I was totally confused by the mismatch between the plots of Figs. 9a, 9e, 9i, 9k and the first line of the caption, "Probability density distribution functions (sum of probabilities corresponding to 1 for each Naero or AOD value)". Figures 11 and 12 also have the same problem. The numbers next to the color bars in these plots are much larger than 1, which is clearly inconsistent with the caption that the sum is 1. Also, these plots of Figs. 9a, 9e, 9i, 9k are collocated with the other plots in which the color contours show a different type of variable, not probability, which further confused me. Finally, I came to the conclusion that the color contours in Figures 9a, 9e, 9i, 9k would show the probability distribution in percentage (%) format. However, this information is not included in either the plots or the captions.

Another big problem is the phrase "the first two lines" and "the third line" in the caption and in the text of the manuscript. I could not understand which line(s) in the series of plots corresponded to "the first two lines" and "the third line". Each plot panel has only one line. Finally, I came to the conclusion that "the first two lines" would mean the lines in Figs. 9a, 9b, 9c, 9d, 9e, 9f, 9g, and 9h, and "the third line" would mean the lines in Figs. 9i, 9j, 9k, and 9l. But this wording is rather ambiguous. If my conclusion above is correct, the authors should have explicitly indicated which lines of which plot panels, or at least these words should have been "the lines in the plots in the first and second rows in Fig. 9" and "the lines in the plots in the third row in Fig. 9".

The position of the plot panels is also quite misleading and unfriendly to the reader. If Figure 9k is in the same series plot as Figures 9a, 9e, and 9i, it should be in the same row or

column of the plot group and follow Figure 9i. Same for Fig. 9l. And why does Fig. 9 have no plots for the relationship between AOD and Nd vs. supersaturation and QV, even though the same plots are included but against the aerosol mean radius?

3. Causality between AOD and precipitation

The discussion in the manuscript does not take into account that the aerosol-cloud interactions are not one-way, especially when precipitation is involved. Figure 3 shows that there is a negative correlation between accumulated precipitation amounts and near-surface concentrations of PM2.5. The constraint of the meteorological variables by the assimilation does not affect the emission from the land surface, except for dust. Thus, the accumulated precipitation amounts were changed by the constraint of meteorological variables using the assimilation as the first order. The change in the accumulated precipitation amounts modified the aerosol concentrations by the wet deposition process. Then, the modified aerosol concentrations can affect the precipitation amounts through the change in cloud microphysical properties as the second order. Therefore, the effect of wet deposition by precipitation is not negligible in the current simulations.

The discussion in Section 3.4 is sometimes misleading regarding the causality between AOD and precipitation, as noted in one specific comment. If the authors believe that this part, which I consider misleading, is correct, please provide evidence through additional levels of analysis to show its validity.

4. The relationship between the multiple mechanisms for cloud development and Nd

There is a conclusion like " multiple supersaturation pathways and abundant aerosols in EC cause Nd to exhibit a much robust increasing trend compared to ECO at low Naero and strong fluctuations at high Naero" (line 453). I agree that the role of abundant aerosols in EC is important, while the role of multiple supersaturation pathways, i.e., multiple mechanisms for cloud development, is still unclear to me. The manuscript does not show the contribution of each mechanism or pathway to the formation of the Nd statistics. If the authors think this is important, please provide additional evidence.

**Specific comments:**

Line 122: As I look at the time series plots of near-surface PM2.5 concentrations at the ground sites in Fig. 4, removing the first 24 hours from the analysis as a spin-up seems insufficient, because the simulation clearly underpredicts the concentrations at most sites in a few days from

the beginning compared to those at later dates in the period. Should at least the first 48 or 72 hours be removed from the analysis?

Line 123: For the initial and lateral boundary conditions of the gas and aerosol species, were any data sets used or not?

Table 1. Both RRTMG and CAM have their own radiation schemes for both long and short wavelengths. Why did the authors use different packages for the long and short wavelengths?

Line 150: Did the four-dimensional data assimilation (FDDA) in WRF apply only to the simulation in the parent domain with 12 km grid spacing (Fig. 1), or also to the simulations in the nested domains with 4 km grid spacing?

Line. 156: Relative humidity is not included in the data sets (relative humidity is already a function of pressure, temperature, and dew point).

Line 160: Please use the correct full name of the product, the Integrated Multi-satellitE Retrievals for GPM (IMERG). Also, please do not forget to include the following citation in the reference as shown on the web page.

Huffman, G.J., E.F. Stocker, D.T. Bolvin, E.J. Nelkin, Jackson Tan (2019), GPM IMERG Final Precipitation L3 1 day 0.1 degree x 0.1 degree V06, Edited by Andrey Savtchenko, Greenbelt, MD, Goddard Earth Sciences Data and Information Services Center (GES DISC), Accessed: [Data Access Date], 10.5067/GPM/IMERGDF/DAY/06

Line 168: Please add the following citation for MOD06_L2.

Platnick, S., Ackerman, S., King, M., et al., 2015. MODIS Atmosphere L2 Cloud Product (06_L2). NASA MODIS Adaptive Processing System, Goddard Space Flight Center, USA: http://dx.doi.org/10.5067/MODIS/MOD06_L2.061

Line 222: There is no citation and data source information for the MODIS AOD in Section 2.3. And is the simulated AOD a "clear sky" AOD, correct?

Line 264: The phrase "supersaturation pathway" is often used in this manuscript, but this is not popular in the community; at least I did a Google search, but nothing came up for atmospheric research. Please add an explanation of the meaning at this first appearance.

Figure 6. These values are a variable (x, y, z, t) in the model. Please include an explanation of how the authors sampled and calculated the values shown in Fig. 6 from the model output.

Figure 8 and Line 291-368. First of all, it is problematic that there is no explanation of how the vertical cross section of the supersaturation was calculated in Figs. 8e and 8f. Since the shown supersaturation is greater than or equal to zero, I assume that the authors only sampled grid points with supersaturation > 0% (i.e., relative humidity > 100%) and plotted the average there. However, if this is the case, the higher supersaturation in Figs. 8e and 8f does not always mean

that supersaturation and clouds occur more frequently, which is probably what the authors would argue around line 296-368.

Lines 298-303: Aren't (2) and (4) identical after all?

Line 351: "temperature and water vapor changes at that time compared to the last time". I do not understand what this means.

Lines 352-359: The plots in Fig. 10 show a ratio of Nd to Naero, not a ratio of locally activated Nd from aerosols to Naero. Since Nd is advected by wind, the existence of Nd is not identical to the existence of aerosol activation.

Lines 360-365: I cannot follow what the authors would argue here. Since supersaturation, i.e., relative humidity, is a function of air pressure, temperature, and water vapor (mixing ratio or specific humidity), the existence of saturation is based simply on whether these three components meet the condition.

Line 385: In the range of Nd above 16,000 cm-3 the sample volume seems to be insufficient to calculate the means of the CLWC for a robust conclusion. Same as in line 388 for non-precipitating clouds. These are inconsistent with the concept of statistical analysis described in line 318, " avoid fluctuations from individual extreme processes and obtain the average state of the liquid phase clouds".

Lines 398-399: The causality in this sentence for Fig. 12d does not make sense to me. As shown in Fig. 3, precipitation has a significant effect on the simulated aerosol concentration through wet deposition. Thus, simulated AOD is low where there is heavy precipitation. Also, high CLWP is needed to develop heavy precipitation.

Line 408: However, each line graph in Figs. 12e and 12h shows a peak near AOD=0.

Line 480: Please submit namelist files of the WRF model simulations in the supplemental materials.

**Grammatical problems:**

"Strong aerosol activation", this or similar wording is often found throughout the manuscript, but is ambiguous. Please rephrase to avoid misunderstanding.

Line 402: "ADO" => "AOD"

---

## Referee Comment (RC2)

Review comments of "Exploring aerosol-cloud interactions in liquid-phase clouds over eastern China and its adjacent ocean using the WRF-Chem-SBM Model" by Zhao et al., 2023

General comments

The authors simulate the liquid-phase clouds in eastern China over land and ocean and explore the different aerosol-cloud processes including aerosol activation, precipitation and entrainment-evaporation in eastern China (EC) and eastern China ocean (ECO). Their simulations use the technical as detailed as possible and evaluation is very valuable. The analysis of aerosol-cloud interactions provides many new insights in this specific region, such as which mechanisms dominant in which region. Overall, I recommend publish it after address some specific comments below. Given that the comments below are mainly at the aspect of presenting and discussion, I guess the reviewer can address them in 2-3 weeks. So I recommend minor revision.

Specific comments

1. The simulation uses SBM, 4d data assimilation and WRF Chem. All those techniques are the current "most" detailed representation of aerosol-cloud interactions. So I believe readers may be curious about the computational cost of this kind of simulations. I think it valuable to describe the computational cost in the method section for other people to decide on their model configurations.

2. "supersaturation pathway", this terminology is mentioned without a clear definition. Based on the content, I guess the "multiple supersaturation pathway" means the multiple contributing factors to supersaturation, or multiple aerosol-cloud processes that impacts supersaturation, is that right? I suggest a clear definition of it. If this terminology was used in previous literature, I recommend citing the papers. For me, "pathway" is usually used to describe the spatial trajectories.

3. Abstract: surface longwave radiative forcing cooling is mentioned. Also, the cloud top radiative cooling is also mentioned in the results section. Please specify which cooling you refer to in the item 3 for EC and item 2 for ECO.

4. Line 174-175: add "respectively" at the end of the sentence

5. Line 185-186: How did you match that? Please clarify it.

6. Figure 2: 4d data assimilation has large effects on temperature and humidity. Are those the two major variables assimilated? Does the assimilation take care of wind also?

7. Figure 4: I don't mind the figure goes in the current form, but add a legend showing the red, blue and black lines would be better.

8. Line 248: "low over land and high over ocean" is only evident for CER, but not Nd. Modify the sentence please.

9. Line 258-259: Aerosol and clouds are still not good. Probably it is better to go through those differences and provide a possible explanation for the differences. I know simulating aerosol and clouds are hard (I believe "everybody" knows that), but it is not good to explain Figure 5 in this way, given the fact that model underestimate CTH, overestimate CTP, overestimate CER over ocean, overestimate Nd over land...

10. Line 263 "produced by anthropogenic emission". To reach this explanation, a plots showing the chemical composition may help. Although Figure 7 can be used to infer this, but a pie diagram is better and clearer.

11. Line 265-266: "ECO aerosols are mainly transported from EC". I can not reach this from Figure 7. Please clarify.

12. Line 315: Two methods are mentioned here. So which way did you use for Figure 9 and 10, and why?

13. Line 325: Do you mean Figure 9a and 9e?

14. Figure 9: The subplots are not the same in size, which looks odd. Also, the order is odd too. Add the four figures in the third row to the end of the first and second row for EC and ECO, respectively. I recommend using log for the x-axis for N_aero and Nc.

15. The authors use the differences between Nd and N_aero in Figure 9, and the ratio of Nd to N_aero in Figure 10. Why did you use different metrics? What would the figure be like if you use another metric?

16. Figure 10: use log for x-axis

17. Figure 10 caption use "water vapor variation", but y-axis use "water vapor changes". Please use consistent word.

18. Line 382: did you use the exact 0? Or a very small threshold values?

19. Line 383-384: this phenomenon is not specific to precipitating clouds. The non-precipitating clouds also has similar trends.

20. Line 384: influence -> net influence

21. The major problem with figure 11 and related text: data after a large Nd may have very small sample size. So I don't think it is valid to derive any conclusion using that portion of plots, say when Nd>16000 cm-3 for EC and >12000 cm-3 for ECO.

---

## Author Comment (AC1)

**Response to the Comments of Referees**

**Journal:** Atmospheric Chemistry and Physics
**Manuscript Number:** egusphere-2023-2858
**Title:** Exploring aerosol-cloud interactions in liquid-phase clouds over eastern China and its adjacent ocean using the WRF-Chem-SBM model
**Author(s):** Jianqi Zhao, Xiaoyan Ma, Johannes Quaas, and Hailing Jia

We thank the reviewers and editor for providing helpful comments to improve the manuscript. We have revised the manuscript according to the comments and suggestions of the referees.
The referee's comments are reproduced (black) along with our replies (blue). All the authors have read the revised manuscript and agreed with the submission in its revised form.

**Anonymous Referee #1**

**Review of "Exploring aerosol-cloud interactions in liquid-phase clouds over eastern China and its adjacent ocean using the WRF-Chem-SBM model" by Zhao et al., submitted to Atmospheric Chemistry and Physics (ACP)**

[Article#: acp-2023-2858]

This report contains general, major, and specific comments from this reviewer on the manuscript.

**A summary of the manuscript and general assessment:**

Recommendation: Reconsidered after major revisions

This study conducted WRF-Chem-SBM simulations to investigate aerosol-cloud interactions, particularly the relationship between aerosol number concentration and cloud droplet number concentration/liquid water content, focusing on the differences between eastern China (EC) and the adjacent ocean (ECO). The simulations reasonably reproduced the distributions of cloud properties and precipitation amounts obtained from satellite measurements, especially when a four-dimensional assimilation was applied. The authors identified dominant mechanisms for cloud development in the EC and ECO. The different relationships between aerosol number concentration and cloud droplet number concentration/liquid water content in EC and ECO are presented and the mechanisms for the differences are discussed.

This study is within the aims and scope of Atmospheric Chemistry and Physics (ACP), specifically the subject for "Aerosols, Cloud and Precipitation", the research activity for "Atmospheric modelling", the altitude range for "Troposphere", and the science focus on "both Chemistry and Physics".

Although I appreciate the authors' efforts in the model simulations using the state-of-the-art model for chemical-aerosol-cloud-precipitation interactions and the analysis, this manuscript has significant problems in readability and presentation, particularly using figures. It was very tough for me to read the whole manuscript and try to understand the contents in the presentation. I found that this manuscript

was resubmitted to ACP after a rejection, and I reviewed the open discussion on the previous manuscript. I have no idea why the previous reviewers did not point out the presentation problems. However, I cannot ignore the problems because they greatly reduce the quality of the manuscript. Also, in my opinion, there are still major problems in the analysis approach of the simulation results and the discussion of the significance of the results with those of other similar studies. More details are provided in the following paragraphs.

First, the presentation using figures has serious problems. The worst figure and caption are in Fig. 9, and I describe my complaints individually in the Major Comments below. The main problem with the figures is that there is little explanation of what is shown in most of the figures and how the values shown are calculated. As a result, almost every time I encountered figures while reading the manuscript, I had to speculate how the authors calculated and produced the plots. This was stressful and confusing. In most cases, I assumed simple spatial and/or temporal averages, not instantaneous values of the variables, because the date and time are not specified in the captions. However, there are some variables, such as positive supersaturation and cloud droplet effective radius (CER), that cannot simply be averaged over all grid points. For example, in Fig. 8, these variables are shown together with Naero and water vapor, which can be simply averaged over all grid points. I am very confused as to whether all variables were averaged over the same sampling or not, yet the authors use the figure to argue for their correlation. In particular, sampling supersaturation needs special care, because it could be negative even in the cloud. The values and the scientific implications of the averaged supersaturation depend heavily on how it is sampled for averaging.

I request the authors to revise the figures, their captions, and the text of the manuscript to explicitly state what is shown and how the values shown in all figures (including Supplementary Materials) were calculated to avoid confusion and misunderstanding by readers. This is one of the minimum requirements for scientific papers.

Thanks the reviewer for the comments and suggestions. We have modified the figures to make them clearer and provided the detailed descriptions in the figure captions and text on how we calculated and produced the plots.

For Fig. 8 mentioned above, the cloud parameters for those non-liquid-phase cloud grids at each time were firstly filtered out when we do the average. The lowest value of supersaturation used in this study is 0 (even if the atmosphere is not saturated). The average value of this supersaturation characterizes the intensity of the supersaturation of EC and ECO during the simulation period. Details can be found in the track-changes version of the manuscript and in responses to specific comments.

When analyzing correlations, we sampled and processed samples that met the conditions for each time point and grid point ($N_d > 1$ cm$^{-3}$, CIWC $= 0$ and supersaturation $> 0$), as detailed in the manuscript and subsequent responses.

Second, the approaches to analyze the simulation results in Sections 3.3 and 3.4 have significant problems. It is difficult for me to tell all the problems, especially related to the statistical analysis, and

how to improve them to the authors in these review comments, but I hope the manuscript will be improved as much as possible for reconsideration. First of all, the authors would perform statistical analysis to obtain the average states without fluctuations from individual extreme processes (line 318). However, Figs. 9-13 clearly show such fluctuations, probably due to the limited sampling volume for the conditions. The parts with fluctuations are reflected in the discussion and conclusions. If the authors want to exclude the effects of individual extreme processes, the conditions with low frequencies should be filtered out in the calculation of the statistics.

Thanks for the suggestions, we have modified the figures and analyses in Sections 3.3 and 3.4. For the problem of "fluctuations" you raised here, we try to filter out the low-frequency conditions, such as if a bin has less than 3 samples, then the bin is set as invalid value. However, this prevents us from analyzing the variations of relevant parameters at high $N_{aero}$ or $N_d$. The inclusion of these low-frequency bins leads to fluctuations at $N_{aero}$ or $N_d$, but allows us to get a better understanding of the overall trend of the relevant parameters. Line 318 in the original manuscript was somewhat inappropriate, and we have replaced it with "the former providing abundant samples and more immediate and detailed aerosol-cloud relationships".

We found some problems with our previous statistical method to the sample of figs. 9-13. Previously, we set 200 (aerosol, x-axis) × 100 ($N_d$, y-axis) bins for the horizontal and vertical coordinate variables of the figure, traversed all samples (each sample contained $N_{aero}$ or AOD, $N_d$, aerosol volume mean radius, supersaturation, and water vapor content values). When the aerosol and $N_d$ values of a sample satisfy the corresponding aerosol and $N_d$ intervals, the value for that sample is the value for that bin. This approach suffers from traversal order, which prevents the complete representation of the aerosol-cloud relationship. In the revised manuscript we corrected the statistical method, it is still the same bin as before, but we first put all the samples that match the aerosol and $N_d$ intervals of each bin into the corresponding bin, and then average the samples in each bin to obtain the value of each bin. We added a note on the method in the caption of Fig.10.

Related to the above problem, there is little information about the probability or frequency of the variables discussed in the sections, except for Nd and CLWP. This makes it difficult to understand how representative each point in the colored contours and lines in Figs.9-13 is of the entire special and temporal sampling volume. For example, Fig. 10a shows a ratio of Nd to Naero averaged over small ranges (bins) of U and Naero, if my understanding is correct. However, this plot does not have the information of how frequent and representative each point is in the whole sample volume. For example, -20 m/s of U is probably a fairly rare case and not representative of what is simulated in EC. Additional levels of statistical analysis for frequency and sampling are needed to unravel the correlation with different weather conditions and finally understand the relationship between Naero and Nd-LWP.

There are some misleading interpretations of the statistical analysis. Please be careful when writing the explanation. For example, a higher mean does not always mean a higher frequency or probability.

We binned each sample according to aerosol, cloud and meteorological parameters (horizontal

and left vertical coordinates in the figure). Each sample contains variables such as $N_{aero}$, $N_d$, CER, supersaturation, water vapor content, etc. The position of the samples in the figure is determined by the intervals represented by these bins, and the values of the probability density distributions in the figure represent the frequency of occurrence of the bin at the corresponding intervals of the horizontal coordinates. We note it in the text and figure captions.

For Figs. 10 and 13 (Figs. 11 and 14 in the revised manuscript), we wanted to analyze the effect of different meteorological and aerosol conditions on aerosol activation and cloud development, and the frequency of samples under that meteorological and aerosol condition is not the focus. And if we focus on the frequency of occurrence of meteorological and aerosol conditions, as shown in the figure below, the vast majority of the samples occur in low $N_{aero}$ conditions, which is not helpful for this study.

We revised the explanations of the statistical analyses in sections 3.3 and 3.4 in accordance with the reviewers' comments, details of which can be found in the track-changes version of the manuscript.

[Figure]

Figure RC1-1. Same as Fig. 11 of the revised manuscript but for number of samples

Third, there is no discussion of what and how the use of the state-of-the-art model improved the simulations of aerosol-cloud interactions compared to other previous modeling studies as well as observations. This is necessary to explicitly demonstrate the scientific significance of this research. For example, as partially described in the introduction of the manuscript, conventional global aerosol transport models have had problems, e.g., they tend to overpredict the increase in LWP in response to increases in Na or Nd, compared to global-scale satellite observations (e.g., Quaas et al. 2009). Did the simulations in this study solve the problem or not? How similar or different is the obtained

relationship between LWP and Na (AOD) or Nd compared to those in high-resolution model simulations in other studies? In addition, there is no direct comparison of the relationship with satellite observations in this case, although the authors sometimes emphasize the advantages in the manuscript. Why was the relationship between LWP and Nd from the simulations not directly compared with that from Terra/Aqua MODIS satellite observations in this case?

Comparisons with other previous aerosol-cloud interaction modeling studies need to be placed on the same scenarios, e.g., same spatial scales, same aerosol and meteorological conditions, and exhaustive comparisons need to be accomplished with a huge amount of workload, which is not the focus of this study. We introduce the advantages of RCM over GCM and the advantages of the bin scheme over the bulk scheme revealed by previous studies in the introduction, and further illustrate the latter in the revised manuscript by citing Zhang et al. (2021b).

Regarding not using the simulation results for direct comparison with MODIS, this is due to the fact that we have too little MODIS available samples due to the limitation of the spatial and temporal scales of our simulations. We interpolated the MODIS data into the model grid so as to match the MODIS cloud data grid with the AOD data grid, and only 36 and 253 samples (we have tens of millions of valid samples from the model) containing both valid cloud parameters and AOD values were obtained at EC and ECO, respectively, and with a small AOD range, which prevented us from obtaining a valid aerosol-cloud relationship. After not requiring samples to contain a valid AOD, we obtained a relatively large number of samples and added a comparison of MODIS with simulated CLWP and $N_d$ relationships before and after assimilation to the revised manuscript (fig. 6).

**Other Major comments:**

1. Caveat when evaluating simulation performance based on RMSE and correlation to the coarser resolution gridded dataset (Fig. 2)

There is a caveat when evaluating the simulation performance based on the RMSE against the coarser resolution gridded data set than the model horizontal grid spacing in Fig. 2. The MICAP is horizontally gridded data with a resolution of 2.5 degrees. This basically means that the data does not contain information about variations at horizontal scales smaller than degrees, which can definitely occur in the real world. Thus, when MICAP is re-gridded into the model horizontal grid structure with finer grid spacing, the interpolated data cannot be fully used as the true value for evaluating spatial variations in the simulated fields containing finer-scale variations represented in the model with finer grid spacing. Therefore, the calculated RMSE for Figure 2 has little validity, and the difference in RMSE calculated in such an approach does not guarantee worse or better performance of the simulations.

From this perspective, the RMSE and correlation against IMERG in Fig. 3 are okay, but those against PM2.5 have the same problem. In the case of PM2.5, the improvement of the mean bias by using the assimilation greatly reduces the RMSE, while it does not guarantee the improvement of

the horizontal variations, especially at smaller scales, as shown by the similar correlation coefficients between the two simulation results.

In order to avoid evaluation uncertainties caused by the fact that low-resolution data do not contain small-scale information, we adjusted the interpolation method to interpolate from high to low resolution when comparing model results to observations. Specifically, MODIS (1-10 km resolution) and IMERG (0.1° resolution) data are interpolated to the WRF grid (12 km resolution) when comparing the model to satellite data, and WRF data are interpolated to the MICAPS grid (2.5° resolution) when comparing the model to MICAPS data. We illustrate it in Section 2.3. In addition, it is somewhat insufficient to assess the impact of assimilation on the meteorological field using only RMSE. We have added the correlation coefficients between the observed and simulated meteorological fields at each vertical layer in Figure 2. In Figure 2, the average values and RMSE compare observations and simulations numerically, and the spatial correlation coefficients compare observations and simulations in terms of spatial distribution, making the evaluation more comprehensive.

Due to the resolution issues described above, we no longer pursue the interpolation of site observation $PM_{2.5}$ data for comparison with the simulation results. Only satellite and observed precipitation and AOD data are retained in Figure 3 to assess the spatial distribution of simulated precipitation and aerosols. Sites with good observational continuity are uniformly selected in the simulation domain to assess the temporal variability of the simulated aerosol (Fig. 4).

2. Figure 9

At first glance, I could not understand what the color contours in these plots were showing at all because of the mismatch between the plots and the caption, as well as the poor wording in the caption. However, after struggling for several hours over days, I finally reached an understanding of what these plots would show to some degree, although I still do not have full confidence. First of all, I was totally confused by the mismatch between the plots of Figs. 9a, 9e, 9i, 9k and the first line of the caption, "Probability density distribution functions (sum of probabilities corresponding to 1 for each Naero or AOD value)". Figures 11 and 12 also have the same problem. The numbers next to the color bars in these plots are much larger than 1, which is clearly inconsistent with the caption that the sum is 1. Also, these plots of Figs. 9a, 9e, 9i, 9k are collocated with the other plots in which the color contours show a different type of variable, not probability, which further confused me. Finally, I came to the conclusion that the color contours in Figures 9a, 9e, 9i, 9k would show the probability distribution in percentage (%) format. However, this information is not included in either the plots or the captions.

Another big problem is the phrase "the first two lines" and "the third line" in the caption and in the text of the manuscript. I could not understand which line(s) in the series of plots corresponded to "the first two lines" and "the third line". Each plot panel has only one line. Finally, I came to the conclusion that "the first two lines" would mean the lines in Figs. 9a, 9b, 9c, 9d, 9e, 9f, 9g, and 9h, and "the third line" would mean the lines in Figs. 9i, 9j, 9k, and 9l. But this wording is rather

ambiguous. If my conclusion above is correct, the authors should have explicitly indicated which lines of which plot panels, or at least these words should have been "the lines in the plots in the first and second rows in Fig. 9" and "the lines in the plots in the third row in Fig. 9".

The position of the plot panels is also quite misleading and unfriendly to the reader. If Figure 9k is in the same series plot as Figures 9a, 9e, and 9i, it should be in the same row or column of the plot group and follow Figure 9i. Same for Fig. 9l. And why does Fig. 9 have no plots for the relationship between AOD and Nd vs. supersaturation and QV, even though the same plots are included but against the aerosol mean radius?

We adjusted the order of the plot panels in Figs. 9 and 12 and changed the captions of most of the images in the manuscript, including Figs. 9-13, to make them easier to understand.

We removed the vague usages such as "the first two lines" in the figure captions, and adopted clearer plot panels orders and figure caption expressions.

We added panels of the relationship between AOD and $N_d$ vs. supersaturation and QV in Fig. 9 (Figs. 10 k-l and o-p of the revised manuscript).

3. Causality between AOD and precipitation

The discussion in the manuscript does not take into account that the aerosol-cloud interactions are not one-way, especially when precipitation is involved. Figure 3 shows that there is a negative correlation between accumulated precipitation amounts and near-surface concentrations of PM2.5. The constraint of the meteorological variables by the assimilation does not affect the emission from the land surface, except for dust. Thus, the accumulated precipitation amounts were changed by the constraint of meteorological variables using the assimilation as the first order. The change in the accumulated precipitation amounts modified the aerosol concentrations by the wet deposition process. Then, the modified aerosol concentrations can affect the precipitation amounts through the change in cloud microphysical properties as the second order. Therefore, the effect of wet deposition by precipitation is not negligible in the current simulations.

The discussion in Section 3.4 is sometimes misleading regarding the causality between AOD and precipitation, as noted in one specific comment. If the authors believe that this part, which I consider misleading, is correct, please provide evidence through additional levels of analysis to show its validity.

We modified the analysis in Section 3.4 and added the statement of bi-directional effects of precipitation and aerosols.

4. The relationship between the multiple mechanisms for cloud development and Nd

There is a conclusion like "multiple supersaturation pathways and abundant aerosols in EC cause Nd to exhibit a much robust increasing trend compared to ECO at low Naero and strong fluctuations at high Naero" (line 453). I agree that the role of abundant aerosols in EC is important, while the role of multiple supersaturation pathways, i.e., multiple mechanisms for cloud

development, is still unclear to me. The manuscript does not show the contribution of each mechanism or pathway to the formation of the Nd statistics. If the authors think this is important, please provide additional evidence.

We have removed the use of "supersaturation pathways" and used more specific descriptions of physical processes (e.g., longwave radiative cooling, terrain uplift, etc.), which are described in this paragraph and in Section 3.2.

**Specific comments:**

Line 122: As I look at the time series plots of near-surface PM2.5 concentrations at the ground sites in Fig. 4, removing the first 24 hours from the analysis as a spin-up seems insufficient, because the simulation clearly underpredicts the concentrations at most sites in a few days from the beginning compared to those at later dates in the period. Should at least the first 48 or 72 hours be removed from the analysis?

This underestimation is caused by chemical initial and boundary conditions rather than insufficient spin-up time (the default chemical initial and boundary conditions were previously used), and it has been resolved by adding initial chemical and boundary conditions from the Community Atmosphere Model with Chemistry (CAM-chem) to the model.

Line 123: For the initial and lateral boundary conditions of the gas and aerosol species, were any data sets used or not?

Previously we used model defaults, and we reran the simulations using initial and boundary data from CAM-chem.

Table 1. Both RRTMG and CAM have their own radiation schemes for both long and short wavelengths. Why did the authors use different packages for the long and short wavelengths?

This is because we need model output cloud water and cloud ice optical thicknesses, and only the CAM and Goddard schemes in WRF calculate these variables. Since the CAM scheme is unable to handle aerosol direct effects, based on the recommendation of the WRF-Chem User's Guide, we re-run the simulation using the RRTMG longwave scheme and the Goddard shortwave scheme to include aerosol direct effects. This configuration is widely used in aerosol-cloud simulations using WRF-Chem (Chapman et al., 2009; Sarangi et al., 2015; Gao et al., 2016).

Reference:

Chapman, E. G., Gustafson Jr, W., Easter, R. C., Barnard, J. C., Ghan, S. J., Pekour, M. S., and Fast, J. D.: Coupling aerosol-cloud-radiative processes in the WRF-Chem model: Investigating the radiative impact of elevated point sources, Atmos. Chem. Phys., 9, 945-964, https://doi.org/10.5194/acp-9-945-2009, 2009.

Gao, W. H., Fan, J. W., Easter, R. C., Yang, Q., Zhao, C., and Ghan, S. J.: Coupling spectral-bin cloud microphysics with the MOSAIC aerosol model in WRF-Chem: Methodology and results for marine stratocumulus clouds, Journal of Advances in Modeling Earth Systems, 8, 1289-1309, https://doi.org/10.1002/2016ms000676, 2016.

Sarangi, C., Tripathi, S., Tripathi, S., and Barth, M. C.: Aerosol-cloud associations over Gangetic Basin during

a typical monsoon depression event using WRF-Chem simulation, J. Geophys. Res.: Atmos., 120, 10,974-910,995, https://doi.org/10.1002/2015JD023634, 2015.

Line 150: Did the four-dimensional data assimilation (FDDA) in WRF apply only to the simulation in the parent domain with 12 km grid spacing (Fig. 1), or also to the simulations in the nested domains with 4 km grid spacing?

The four-dimensional data assimilation approach is used in both parent domain and nested domains, we add this note in section 2.2.

Line. 156: Relative humidity is not included in the data sets (relative humidity is already a function of pressure, temperature, and dew point).

Thanks for the correction, we've revised it.

Line 160: Please use the correct full name of the product, the Integrated Multi-satellitE Retrievals for GPM (IMERG). Also, please do not forget to include the following citation in the reference as shown on the web page.

Huffman, G.J., E.F. Stocker, D.T. Bolvin, E.J. Nelkin, Jackson Tan (2019), GPM IMERG Final Precipitation L3 1 day 0.1 degree x 0.1 degree V06, Edited by Andrey Savtchenko, Greenbelt, MD, Goddard Earth Sciences Data and Information Services Center (GES DISC), Accessed: [Data Access Date], 10.5067/GPM/IMERGDF/DAY/06

We've revised and added the citation in the revised manuscript.

Line 168: Please add the following citation for MOD06_L2.

Platnick, S., Ackerman, S., King, M., et al., 2015. MODIS Atmosphere L2 Cloud Product (06_L2). NASA MODIS Adaptive Processing System, Goddard Space Flight Center, USA: http://dx.doi.org/10.5067/MODIS/MOD06_L2.061

We've added it.

Line 222: There is no citation and data source information for the MODIS AOD in Section 2.3. And is the simulated AOD a "clear sky" AOD, correct?

We've added this information.

In the comparison with the MODIS AOD, the simulated AOD was only temporally and spatially matched to the MODIS valid values, and was not restricted to "clear sky".

Line 264: The phrase "supersaturation pathway" is often used in this manuscript, but this is not popular in the community; at least I did a Google search, but nothing came up for atmospheric research. Please add an explanation of the meaning at this first appearance.

We have deleted the use of "supersaturation pathway" in the text and replaced it with a description of specific physical processes or the use of terms such as "contributing factors to supersaturation" and "processes affecting supersaturation".

Figure 6. These values are a variable (x, y, z, t) in the model. Please include an explanation of how the authors sampled and calculated the values shown in Fig. 6 from the model output.

In order to obtain the values shown in Fig. 6, the aerosols and cloud droplets number concentrations of each bins were first vertically weighted averaged into three-dimensional data containing only time, longitude, and latitude, and only the vertical layers of the liquid-phase cloud were weighted averaged, i.e., the layers with CIWC>0 were excluded from the calculations. Subsequent direct averaging of the three-dimensional number concentrations in each bins obtained the values in Fig. 6. We added this note to the figure caption.

Figure 8 and Line 291-368. First of all, it is problematic that there is no explanation of how the vertical cross section of the supersaturation was calculated in Figs. 8e and 8f. Since the shown supersaturation is greater than or equal to zero, I assume that the authors only sampled grid points with supersaturation > 0% (i.e., relative humidity > 100%) and plotted the average there. However, if this is the case, the higher supersaturation in Figs. 8e and 8f does not always mean that supersaturation and clouds occur more frequently, which is probably what the authors would argue around line 296-368.

The lower limit of supersaturation used in the figure is 0, i.e., when the atmosphere is not saturated, the supersaturation is also 0 rather than negative, and 0 values are also involved in the calculation of the average value. The average value of this supersaturation characterizes the overall intensity of supersaturation in EC and ECO during the simulation period. We added this note to the figure caption.

Lines 298-303: Aren't (2) and (4) identical after all?

(2) Water vapor and temperature changes caused by advection are dominated by the meteorological field and occur in both EC and ECO. (4) Topographic uplift, however, is topographically dominated and occurs only in EC where topographic relief exists.

Line 351: "temperature and water vapor changes at that time compared to the last time". I do not understand what this means.

Since the model outputs once per hour, the changes are the current temperature and water vapor of the sample minus the previous hour's value. We added it in the text.

Lines 352-359: The plots in Fig. 10 show a ratio of Nd to Naero, not a ratio of locally activated Nd from aerosols to Naero. Since Nd is advected by wind, the existence of Nd is not identical to the existence of aerosol activation.

The model for this study integrates once every 20 seconds, but due to limitations in computer storage and processing power, the model can only output once per hour. Between two consecutive model outputs, 180 calculations of processes such as transport, activation, and deposition were performed, preventing us from accurately tracking locally activated $N_d$. In addition, not only $N_d$ but also $N_{aero}$ is synchronized by advective transport by wind, $N_d/N_{aero}$ can characterize overall

activation intensity under different meteorological fields and aerosol conditions. In a similar approach, where accurate tracking is not possible, López-Romero et al. (2021) used $PM_{2.5}/PM_{10}$ to characterize the percentage of anthropogenic aerosols. In addition, we selected more than 40 million samples at each time, grid point, and vertical level, and the statistics of such a large sample can basically characterize its overall characteristics.

Reference:

López-Romero, J. M., Montávez, J. P., Jerez, S., Lorente-Plazas, R., Palacios-Peña, L., and Jiménez-Guerrero, P.: Precipitation response to aerosol–radiation and aerosol–cloud interactions in regional climate simulations over Europe, Atmos. Chem. Phys., 21, 415–430, https://doi.org/10.5194/acp-21-415-2021, 2021.

Lines 360-365: I cannot follow what the authors would argue here. Since supersaturation, i.e., relative humidity, is a function of air pressure, temperature, and water vapor (mixing ratio or specific humidity), the existence of saturation is based simply on whether these three components meet the condition.

There is indeed some redundancy here. We changed it to "high $N_d$ to $N_{aero}$ ratio in EC mainly occurs at low temperatures and low humidity conditions, which is due to the fact that the temperature and humidity horizontal gradients are essentially the same (Fig. S2), and EC with low overall water vapor content is more likely to reach supersaturation at both low temperature and water vapor content, and becomes increasingly difficult to reach supersaturation when the temperature and humidity are simultaneously increased".

Line 385: In the range of Nd above 16,000 cm-3 the sample volume seems to be insufficient to calculate the means of the CLWC for a robust conclusion. Same as in line 388 for non-precipitating clouds. These are inconsistent with the concept of statistical analysis described in line 318, " avoid fluctuations from individual extreme processes and obtain the average state of the liquid phase clouds".

We revised the analysis in this paragraph, and in addition we have adopted a more appropriate statement for line 318 (see the response to the second summary comment for details).

Lines 398-399: The causality in this sentence for Fig. 12d does not make sense to me. As shown in Fig. 3, precipitation has a significant effect on the simulated aerosol concentration through wet deposition. Thus, simulated AOD is low where there is heavy precipitation. Also, high CLWP is needed to develop heavy precipitation.

We modified this analysis and added the statement of bi-directional effects of precipitation and aerosols.

Line 408: However, each line graph in Figs. 12e and 12h shows a peak near AOD=0.

As in the above response, we modified the paragraph.

Line 480: Please submit namelist files of the WRF model simulations in the supplemental materials.

We submitted the namelist file of the model simulation in the supplement file (Supplement A).

**Grammatical problems:**

"Strong aerosol activation", this or similar wording is often found throughout the manuscript, but is ambiguous. Please rephrase to avoid misunderstanding.

We replaced it with "high $N_d$ to $N_{aero}$ ratio" and other clearer statements.

Line 402: "ADO" => "AOD"

Corrected.

---

## Author Comment (AC2)

**Response to the Comments of Referees**

**Journal:** Atmospheric Chemistry and Physics
**Manuscript Number:** egusphere-2023-2858
**Title:** Exploring aerosol-cloud interactions in liquid-phase clouds over eastern China and its adjacent ocean using the WRF-Chem-SBM model
**Author(s):** Jianqi Zhao, Xiaoyan Ma, Johannes Quaas, and Hailing Jia

We thank the reviewers and editor for providing helpful comments to improve the manuscript. We have revised the manuscript according to the comments and suggestions of the referees.
The referee's comments are reproduced (black) along with our replies (blue). All the authors have read the revised manuscript and agreed with the submission in its revised form.

**Anonymous Referee #2**

**Review comments of "Exploring aerosol-cloud interactions in liquid-phase clouds over eastern China and its adjacent ocean using the WRF-Chem-SBM Model" by Zhao et al., 2023**

**General comments**

The authors simulate the liquid-phase clouds in eastern China over land and ocean and explore the different aerosol-cloud processes including aerosol activation, precipitation and entrainment-evaporation in eastern China (EC) and eastern China ocean (ECO). Their simulations use the technical as detailed as possible and evaluation is very valuable. The analysis of aerosol-cloud interactions provides many new insights in this specific region, such as which mechanisms dominant in which region. Overall, I recommend publish it after address some specific comments below. Given that the comments below are mainly at the aspect of presenting and discussion, I guess the reviewer can address them in 2-3 weeks. So I recommend minor revision.

**Specific comments**

1. The simulation uses SBM, 4d data assimilation and WRF Chem. All those techniques are the current "most" detailed representation of aerosol-cloud interactions. So I believe readers may be curious about the computational cost of this kind of simulations. I think it valuable to describe the computational cost in the method section for other people to decide on their model configurations.

   Thanks for suggestions. Using the model configurations of this study, EC and ECO simulations require around 15,000 and 10,000 CUP core-hours, respectively. We have added this information in section 2.1.

2. "supersaturation pathway", this terminology is mentioned without a clear definition. Based on the content, I guess the "multiple supersaturation pathway" means the multiple contributing factors to supersaturation, or multiple aerosol-cloud processes that impacts supersaturation, is that right? I suggest a clear definition of it. If this terminology was used in previous literature, I recommend citing the papers. For me, "pathway" is usually used to describe the spatial trajectories.

We have deleted the use of "supersaturation pathway" in the text and replaced it with a description of specific physical processes or the use of terms such as "contributing factors to supersaturation" and "processes affecting supersaturation".

3. Abstract: surface longwave radiative forcing cooling is mentioned. Also, the cloud top radiative cooling is also mentioned in the results section. Please specify which cooling you refer to in the item 3 for EC and item 2 for ECO.

Both the abstract and section 3.3 discuss the effect of meteorological fields on aerosol-cloud, where cooling refers to cooling due to cold northerly winds. We revised the abstract and the text.

4. Line 174-175: add "respectively" at the end of the sentence

We've added it.

5. Line 185-186: How did you match that? Please clarify it.

In order to compare the WRF simulation with MODIS, we first interpolated the MODIS data to the WRF grid to make the coordinates match, for each grid and each time the simulated value is available for analysis only when the MODIS data is valid, otherwise the simulated value is set as the missing value and does not participate in the calculation. We have added in Section 2.3 of the modified manuscript.

6. Figure 2: 4d data assimilation has large effects on temperature and humidity. Are those the two major variables assimilated? Does the assimilation take care of wind also?

Temperature and wind is assimilated directly, and humidity is indirectly affected by assimilating temperature, dew point, wind and air pressure. We provide this clarification at section 2.2 and the beginning of the second paragraph of section 3.1.

7. Figure 4: I don't mind the figure goes in the current form, but add a legend showing the red, blue and black lines would be better.

We added a legend to Figure 4a, along with the figure caption indicating what each color line represents.

8. Line 248: "low over land and high over ocean" is only evident for CER, but not Nd. Modify the sentence please.

We have modified this paragraph.

9. Line 258-259: Aerosol and clouds are still not good. Probably it is better to go through those differences and provide a possible explanation for the differences. I know simulating aerosol and clouds are hard (I believe "everybody" knows that), but it is not good to explain Figure 5 in this way, given the fact that model underestimate CTH, overestimate CTP, overestimate CER over ocean, overestimate Nd over land...

   We have modified the description in Section 3.1 to point out in more detail on overestimation and underestimation of the variables.

   In addition, we effectively improved the model's ability to simulate aerosols by providing the model with chemical initial and boundary conditions from Community Atmosphere Model with Chemistry (the WRF defaults were previously used).

10. Line 263 "produced by anthropogenic emission". To reach this explanation, a plots showing the chemical composition may help. Although Figure 7 can be used to infer this, but a pie diagram is better and clearer.

    Thanks for suggestion, we replaced the original Figure 7 with a pie diagram (Figure 8 in the revised manuscript) to show the results more clear.

11. Line 265-266: "ECO aerosols are mainly transported from EC". I can not reach this from Figure 7. Please clarify.

    This is shown in Fig. 8b of the revised version, i.e. ECO's locally emitted chloride and sodium aerosols contribute less than 20% of the total aerosol mass.

12. Line 315: Two methods are mentioned here. So which way did you use for Figure 9 and 10, and why?

    Fig. 9 (Fig. 10 in the revised manuscript) shows the results by sampling from each vertical layer (left two columns) and the sampling of the entire columns (right two columns). In Fig. 10 (Fig. 11 in the modified manuscript), we used the sampling data from each vertical layer in order to reflect the aerosol-cloud-meteorological field relationship in a more detailed and immediate way. We added notes in Fig. 9, the figure captions for Figs. 9 and 10, and in the main text of Section 3.3.

    In addition, we found some problems with our previous statistical method to the sample of figs. 9-13. Previously, we set 200 (aerosol, x-axis) × 100 (Nd, y-axis) bins for the horizontal and vertical coordinate variables of the figure, traversed all samples (each sample contained Naero or AOD, Nd, aerosol volume mean radius, supersaturation, and water vapor content values). When the aerosol and Nd values of a sample satisfy the corresponding aerosol and Nd intervals, the value for that sample is the value for that bin. This approach suffers from traversal order, which prevents the

complete representation of the aerosol-cloud relationship. In the revised manuscript we corrected the statistical method , it is still the same bin as before, but we first put all the samples that match the aerosol and Nd intervals of each bin into the corresponding bin, and then average the samples in each bin to obtain the value of each bin. We added a note on the method in the caption of Fig.10.

13. Line 325: Do you mean Figure 9a and 9e?

Yes, we corrected it.

14. Figure 9: The subplots are not the same in size, which looks odd. Also, the order is odd too. Add the four figures in the third row to the end of the first and second row for EC and ECO, respectively. I recommend using log for the x-axis for N_aero and Nc.

We adjusted the order and size of the subplots.

The figure below shows the result of using $N_{aero}$ and $N_d$ logarithmic coordinates, it causes the figure to overemphasise high values at low $N_{aero}$ and $N_d$, which is not conducive to showing the overall change. So we still use the original coordinates.

[Figure]

**Figure RC2-1.** Same as Fig. 10 of the revised manuscript, but using log for the x-axis for $N_{aero}$ and $N_d$

15. The authors use the differences between Nd and N_aero in Figure 9, and the ratio of Nd to N_aero in Figure 10. Why did you use different metrics? What would the figure be like if you use another metric?

Using the ratio of $N_d$ and $N_{aero}$ allows us to see the strength of aerosol activation under different meteorological fields and aerosol conditions, whereas the direct use of $N_d$ values as shown in the

figure below, whose high values are overall skewed towards the high $N_{aero}$ coordinates, is not conducive to our understanding of the effect of different aerosol conditions on activation.

[Figure]

**Figure RC2-2.** Same as Fig. 11 of the revised manuscript but for $N_d$

16. Figure 10: use log for x-axis

Shown below, as before, using log for x-axis would make the figure overemphasize information about high values at low $N_{aero}$, which is not conducive to showing the overall variation, so we used the original coordinates.

[Figure]

**Figure RC2-3.** Same as Fig. 11 of the revised manuscript, but using log for x-axis

17. Figure 10 caption use "water vapor variation", but y-axis use "water vapor changes". Please use consistent word.

It should be "change", we revised the figure captions.

18. Line 382: did you use the exact 0? Or a very small threshold values?

The original was exactly 0, but this led to too few samples of non-precipitating clouds and difficulty in clearly distinguishing the difference between precipitating and non-precipitating clouds, so we used new filtering criteria for precipitating (rainwater content above 1 mg·m-3 for each vertical layer and above 1 g·m-2 for column) and non-precipitating (rainwater content below 0.001 mg·m-3 for each vertical layer and below 0.001 g·m-2 for column) clouds in the revised manuscript.

19. Line 383-384: this phenomenon is not specific to precipitating clouds. The non-precipitating clouds also has similar trends.

We re-analyzed it using more appropriate sampling and statistical methods (see response to comment 12 for details).

20. Line 384: influence -> net influence

We've revised it.

21. The major problem with figure 11 and related text: data after a large Nd may have very small sample size. So I don't think it is valid to derive any conclusion using that portion of plots, say when Nd>16000 cm-3 for EC and >12000 cm-3 for ECO.

We revised these texts by replacing them with statements such as "after $N_d$ reaches its peak" and "the near-surface with high aerosol concentrations"…

---

## Author Response (AR2)

**Response to the Comments of Referee**

**Journal:** Atmospheric Chemistry and Physics
**Manuscript Number:** egusphere-2023-2858
**Title:** Exploring aerosol-cloud interactions in liquid-phase clouds over eastern China and its adjacent ocean using the WRF-Chem-SBM model
**Author(s):** Jianqi Zhao, Xiaoyan Ma, Johannes Quaas, and Hailing Jia

We thank the reviewer and editor for providing helpful comments to improve the manuscript. We have revised the manuscript according to the comments and suggestions of the referee.

The referee's comments are reproduced (black) along with our replies (blue). All the authors have read the revised manuscript and agreed with the submission in its revised form.

**Anonymous Referee #1**

**Review of "Exploring aerosol-cloud interactions in liquid-phase clouds over eastern China and its adjacent ocean using the WRF-Chem-SBM model" by Zhao et al., submitted to Atmospheric Chemistry and Physics (ACP)**

**[Article#: acp-2023-2858-version2]**

**Recommendation: Minor revisions (technical corrections)**

Overall, the authors have done great work to address the problems in the presentation that I identified in the previous review. The responses and actions to my second and third major concerns in my general assessment comment in the previous review are not excellent, but acceptable. At this stage of the review, I suggest minor revisions for the revised manuscript. Once the issues listed below have been addressed, I believe the manuscript can be accepted for publication in ACP.

Table 1. The parameterization for the shortwave radiation was changed from CAM to Goddard in the revision. But the citation is still Zhong et al. (2016) for CAM. Please correct the citation.

The earliest version had a reference to (Collins et al., 2004) for CAM, and after changing CAM to Goddard, we changed the reference to (Zhong et al., 2016). The change to this citation is not reflected in the track-changes version of the manuscript due to the literature management software we used.

Fig. 8. Why are the Chloride, Sodium, and Primary Organics portions separated from the main body of the pie chart?

This is due to the default settings of the drawing scripts we use. This can be somewhat misleading and we have revised it.

Line 141: "CUP" => "CPU"

Corrected.

Line 275: "hinder" => "hinders"

Corrected.

Line 359. "temperature and water vapor changes at that time compared to the last time (the model outputs once per hour, the changes are the values of the current time minus the previous hour)" If my understanding is correct, this should simply be described as "changes in temperature and water vapor per hour".

Thanks for the suggestion, we have revised it.

Below are some of my remaining concerns with the revised manuscript. Although the authors do not need to address these, I leave this for possible open discussion.

The authors now clearly describe how to calculate averaged supersaturation, where unsaturated grid points are included as a value of 0 for the average. The definition is fine, but I am not fully convinced that the interpretation of the results in the manuscript is correct, although the interpretation does not affect the conclusions so much.

In parts of Sections 3.3 and 3.4 on statistical analysis, I feel that the discussion of the causality between each parameter is still not well organized, the interpretation may be misleading, and some conclusions are not fully supported. However, the results presented using the state-of-the-art model are valuable to the community.

Thank you very much for your comments. The explanation for supersaturation is as relatively reasonable as we can come up with for that average variation. However, averaging over such a diverse set of samples may obscure some more nuanced processes and relationships. There is also significant room for improvement in our analytical methods. In our future research, we will attempt to specifically track a particular process and enhance our analytical approach to gain a clearer understanding of the relationship, aiming to provide a more reasonable explanation.

---

## Author Response (AR3)

**Public justification (visible to the public if the article is accepted and published):**
Dear Authors,

The abstract is currently too long. Please rewrite it following
https://www.atmospheric-chemistry-and-physics.net/policies/guidelines_for_authors.html

Xiaohong Liu

Dear Editor,

Thank you very much, we have rewritten the abstract to make it meet the requirements.

Jianqi Zhao and co-authors